

**Intra-seasonal hydrological processes on the western Tibetan Plateau: Monsoonal**
**and convective rainfall events ~7.5 ka ago.**
Linda Taft[a,*], Uwe Wiechert[b], Christian Albrecht[c], Christian Leipe[b], Sumiko Tsukamoto[d], Thomas
Wilke[c], Hucai Zhang[e], Frank Riedel[b]
[a]*Department of Geography, University of Bonn, Germany*
[b]*Institute of Geological Sciences, Freie Universität Berlin, Germany*
[c]*Department of Animal Ecology and Systematics, Justus Liebig University Giessen, Germany*
[d]*Leibniz Institute for Applied Geophysics, Hannover, Germany*
[e]*Institute of Plateau Lake Ecology and Pollution Management, Yunnan University, China*
*Corresponding author. E-mail address: ltaft@uni-bonn.de
Keywords
Early Middle Holocene
Bangong Co
Molluscs
Stable isotopes
**Abstract**
Billions of people depend on the precipitation of the Asian monsoons. The Tibetan Plateau and the
Himalayas on the one hand strongly influence the monsoonal circulation pattern and on the other hand
represent water towers of humanity. Understanding the dynamics of the Asian monsoons is one of the
prime targets in climate research. Modern coupling of atmospheric circulation and hydrological cycle
over and on the plateau can be observed and outlined, and lake level controlling factors be identified.



Recent monitoring of lakes showed that many of them have grown at least for decades, the causes
being higher meltwater inflow or stronger rainfall of different sources, depending on the particular
location of a drainage basin. The long-term dynamics, however, can be described best with the aid of
high-resolution climate archives. We focus here on the often controversial discussion of Holocene lake
development and selected the Bangong Co drainage basin on the western Tibetan Plateau as a case
site. The aim of our study is, to identify the factors influencing lake level such as monsoonal or
convective precipitation and meltwater. For doing so, shells of the aquatic gastropod genus *Radix* were
collected from an early Middle Holocene sediment sequence in the Nama Chu sub-catchment of the
eastern Bangong Co and sclerochronlogical isotope patterns of five shells obtained in weekly to sub-
monthly resolution. Our data suggests that during ca. 7.5 ka ago, monsoonal rainfall was higher than
today. However, summer precipitation was not continuous but affected the area as extended moisture
pulses. This implicates that the northern boundary of the SW Asian monsoon was similar to modern
times. We could identify convective rainfall events significantly stronger than today. We relate this to
higher soil moisture and larger lake surface areas under higher insolation. The regional meltwater
amount corresponds with westerly-derived winter snowfall. The snowfall amount was probably similar
to modern times. Exceptionally heavy $\delta^{13}C$ values archived in the shells were likely, at least partly,
triggered by biogenic methane production. We suggest that our approach is suitable to study other lake
systems on the Tibetan Plateau from which fossil *Radix* shells can be obtained. It may thus help to
infer palaeo-weather patterns across the plateau.

**1. Introduction**

*1.1. Background and scope*

The importance of the Tibetan Plateau and the Himalayas for the Asian atmospheric circulation
patterns, particularly their influence on Asian monsoon intensities and distributions, has been
demonstrated in numerous studies (e.g. Harris, 2006; Molnar et al., 2009; Boos and Kuang, 2010,
Chen et al., 2010). The area represents a water tower furnishing large regions of eastern and southern



Asia (Immerzeel et al., 2010; Jacob et al., 2012), and it thus is of major interest to better understand
the coupling of atmospheric circulation and the hydrological cycle. Various lake systems on the
Tibetan Plateau have been studied with particular focus on Late Glacial and Holocene lake level
fluctuations evidencing changes in the hydrological cycle (e.g. Van Campo and Gasse, 1993; Avouac
et al., 1996; Lehmkuhl and Haselein, 2000; Ahlborn et al., 2015; Shi et al., 2017; Wünnemann et al.,
2018). Accordingly, since the Late Glacial lakes on the plateau were largest during the Early and
Middle Holocene and shrank to modern size during the Late Holocene (e.g. Lee et al., 2009; Liu et al.,
2013; Shi et al., 2017; in respect of the formal subdivision of the Holocene we follow Walker et al.,
2012). In modern times the numerous lakes scattered on the plateau span an area of 30,000 to 50,000
km² (Zheng, 1997; Kong et al., 2007; Ma et al., 2011) but this area was up to four times larger during
the Early and Middle Holocene (Hudson and Quade, 2013; Liu et al., 2013).

Recent, mainly satellite-based monitoring suggested that several lakes on the Tibetan Plateau

have grown at least since the 1970s (e.g. Liu et al., 2009; Zhang et al., 2011; Lei et al., 2013; Clewing
et al., 2014a). In respect of the western Tibetan Plateau, including Ladakh, Hutchinson (1937)
concluded that the contemporary lake level rise started already during the late 19th century. Causes
might be increase of meltwater inflow (Zhang et al., 2011) or higher monsoonal and/or westerly-
derived precipitation (Lei et al., 2013), depending on the particular lake system and its position on the
plateau. Kurita and Yamada (2008) discussed the role of local moisture recycling for the precipitation
amount and found it significant for the central Tibetan Plateau. The same hydrological factors of lake
dynamics on the Tibetan Plateau have to be considered throughout the Holocene (e.g. Gasse et al.,
1991; Wünnemann et al., 2010; Bird et al., 2014; Hou et al., 2017). However, they are often
controversially discussed against the background that palaeo-moisture sources have to be
reconstructed using proxies of different quality (e.g. Taft et al., 2014; Hillman et al., 2017;
Wünnemann et al., 2018).

One promising avenue of research to identify palaeo-hydrological processes is the

interpretation of stable isotope ratios of carbonatic lake sediments or corresponding carbonate shells
(e.g. Mischke et al., 2005; Henderson et al., 2010; Qiang et al., 2017; Liu et al., 2018). Observations of
modern processes and analyses of stable isotope behavior in precipitation, rivers and lakes have



provided a solid fundament for the interpretation of Tibetan Plateau palaeo-data retrieved from proxies
(e.g. Araguás-Araguás, 1998; Pande et al., 2000; Gajurel et al., 2006; Tian, 2007; Hren et al., 2009;
Bershaw et al., 2012; Taft et al., 2012; Yao, 2013; Gao, 2014; Mishra et al., 2014; Biggs et al., 2015;
He et al., 2015).

It is under discussion whether plateau lakes had an extension during the Middle Holocene

similar to that of the Early Holocene (e.g. Liu et al., 2013; Ahlborn et al., 2015; Shi et al., 2017), and
when the climate was warmest and most humid (e.g. Morrill et al., 2006; Cheung et al., 2014). There
are several studies, however, indicating that the factors controlling lake level may have changed prior
to the Late Holocene aridification (e.g. Wei and Gasse, 1999; Bird et al., 2014; Shi et al., 2017). In
addition, monsoonal precipitation across the Tibetan Plateau is triggered by SW and SE Asian summer
monsoons on the eastern to central plateau while the western plateau is solely influenced by the SW
Asian monsoon (e.g. Chen et al., 2015; Ramisch et al., 2016; Wünnemann et al., 2018). Controversial
conclusions about the timing of humidity and temperature changes are also due to asynchronous
behavior of the different Asian monsoon branches (e.g. An et al., 2000; Wang et al., 2010; Hudson and
Quade, 2013). The Holocene climate optimum period was likely earlier on the western plateau than on
the eastern plateau (e.g. Wang et al., 2010; Chen et al., 2015).

For Bangong Co, the largest lake system on the western Tibetan Plateau, Gasse et al. (1996)

attributed the Holocene lake level changes mainly to changes of the SW Asian summer monsoon.
Correspondingly, highest lake levels were assigned to monsoonal moisture maxima during ~9.5 to 8.7
cal. ka B.P. and ~7.2.-6.3 cal. ka B.P. (Fontes et al., 1996; Gasse et al., 1996). Kong et al. (2007),
however, concluded that even an enhanced SW Asian monsoon during the Early Holocene did not
affect the western Tibetan Plateau significantly. Kong et al. (2007) though referred to Sumxi Co (Fig.
1), a lake system ca. 120 km north of Bangong Co and thus farther from the northernmost monsoon
front. Based on a cosmogenic $^{10}$Be chronology of the palaeo-shorelines, the authors summarized that
high lake levels were most likely associated with increased recharge from melting glaciers.
Wünnemann et al. (2010) reported for the neighboring Tso Kar (Fig. 1) that monsoonal precipitation
was at maximum from 11.5 to 8.6 cal. ka B.P. but highest lake levels occurred during the early Middle
Holocene due to meltwater increase. In respect of nearby Tso Moriri (Fig. 1), Leipe et al. (2014) also



suggested that meltwater was the main source to increase the lake level during the Middle Holocene
but considered convective rainfall. As mentioned earlier, modern land-atmospheric moisture recycling
is known from the Tibetan Plateau (Kurita and Yamada, 2008) and e.g. short-term convective rainfall
has been observed over western Tibet and Ladakh (e.g. Gasse et al., 1991; Fontes et al., 1993; personal
observations), however, not yet been inferred from Holocene proxies.

The aim of our study is to infer intra-seasonal hydrological processes on the Tibetan Plateau

during the Middle Holocene, using Bangong Co as a model site. Which moisture sources were
significant for the lake dynamics? How can we differentiate between monsoonal, westerly-derived or
convective precipitation and meltwater? Can we distinguish regional convective from monsoonal
rainfall, which both occur during the summer months? A potentially suitable, intra-seasonal
environmental archive, which is available across the plateau, are the shells of the aquatic gastropod
*Radix* (Basommatophora, Lymnaeidae). Taft et al. (2012, 2013) demonstrated that sclerochronological
stable isotope patterns from *Radix* shells allow to outline hydrological processes in a sub-monthly
resolution.

*1.2. Regional setting and study areas*

All study sites are located within the Bangong Co drainage basin (Figs. 1 and 2). The basin contains
five interconnected lake sub-basins forming the transboundary Bangong Co lake system at 4241 m
a.s.l. (SRTM elevation data v4.1; Fig. 2). It comprised a total water surface area of ca. 611 km² in
2012, and stretches ca. 160 km within the western Bangong suture zone (Fig. 1; Fontes et al., 1996,
Dortch et al., 2011; Gourbet et al., 2017). Particularly in the eastern Bangong Co lake system, a
number of palaeo-shoreline features were observed (e.g. Fontes et al., 1996; Dortch et al., 2011;
Clewing et al., 2014a; Fig. 3A), some as high as ca. 80 m above contemporary lake level, witnessing
strong past lake level fluctuations and possibly indicating maximum lake extension during the Upper
Pleistocene (Shi et al., 2001; Yu et al., 2001). Dortch et al. (2011) suggested a lake level ca. 10 m
higher than the modern one during the Early to Middle Holocene, which resulted in a lake area of ca.
810 km².




**Figure 1**

The Bangong Co drainage basin spans an area of ca. 31348 km$^2$ (including the lake surface

area). The mountain ranges, which delimit the watershed, exceed 6000 m a.s.l. (Fig. 1). Cretaceous
granodiorites, Cenozoic sandstones and conglomerates, and Late Palaeozoic and Jurassic limestones
are widely distributed (Wang and Hu, 2004; Gourbet et al., 2017). Roughly, two thirds of the total
catchment area drain into the easternmost basin, Nyak Co (Fig. 2). This causes an overspill of Nyak
Co to its neighboring basin. Although Bangong Co has been a closed basin since ca. 7 ka when the
palaeo-outflow Tangtse, a river valley connecting to the western terminus of the lake system (Fig. 1),
was probably active for the last time (Brown et al. 2003; but compare Dortch et al., 2011), only the
westernmost basin behaves like an endorheic lake. Four of the five lakes are overflowing to their
neighboring basin in the west (personal observation FR, 2012). While Nyak Co has a relatively low
salinity of ca. 0.5 psu, salinity increases significantly along the other basins (Ou, 1981; Wilckens,
2014). This is in phase with $\delta^{18}O_w$, which shows a trend to heavier values towards the west (Wilckens,
2014; Wen et al., 2016).

**Figure 2**
**Table 1**

The large alluvial fan of Chiao Ho (Fig. 2A, fossil shells site B) demonstrates long-term low-

frequency activity of the northern Nyak Co catchment, which includes meltwater from glaciers (Wei et
al., 2015). $\delta^{18}O_w$ of Chiao Ho (Fig. 2A, location 6) was ca. -13,9‰ (Wilckens et al., 2014; Table 1).
$\delta^{18}O_w$ from the southern Nyak Co catchment, Makha River (Fig. 2A, location 7) and tributaries, is
similar (Fontes et al., 1996; Wilckens, 2014; Wen et al., 2016; Table 1). The eastern catchment of
Nyak Co is mainly drained by the Nama Chu and its tributaries (Figs. 2 and 3). The Nama Chu sub-
catchment area spans ca. 3420 km$^2$ (Fig. 1). Hydrological parameters vary stronger here than in the



other sub-catchments and are compiled in Table 1 (locations 1-3, in Fig. 2). The Nama Chu and Chiao
Ho sub-catchments are in direct neighborhood (Fig. 1).

**Figure 3**

The Nama Chu valley represents the main study area. Nama Chu represents partly a fluvial

system and partly a sequence of ponds (Fig. 3A). We studied the Nama Chu valley from the river
mouth at Nyak Co upwards to a saline pond (~30 psu), ca. 28 km from Nyak Co (Figs. 3A, B). The
morphology of the valley is controlled by tectonics and alluvial, fluvial and periglacial processes.
Alluvial fans, particularly from northern tributaries, block the water flow at several sections leading to
the formation of ponds (Fig. 3A; personal observations 2009, 2012). It is likely that during past
periods of higher precipitation the general morphology of the fluvio-lacustrine system was similar,
based on the assumption that stronger water flow along the Nama Chu was synchronous to stronger
lateral alluvial transport into the valley (see Fig. 3A). Field investigation (2012) of sediments exposed
in Nama Chu pond basins (e.g. see Fig. 3B) demonstrated that permafrost mounds are widely
distributed. They were probably formed by uplift of pond mud, the high water content of which
became subject to continuous segregated ice formation (Wünnemann et al., 2008) when the mud
became exposed during low water levels.

At the northern edge of the saline pond (triangle in Fig. 2B, camera symbol in Fig. 3A), ca. 4-

5 m above the water level, we found sediments containing fossil shells of aquatic molluscs (Fig. 3C).
This site is located ca. 45 m above the 2012 lake level of Nyak Co and we conclude that Nyak Co
could not capture the palaeo-habitat during its Middle Holocene highstand. Consequently, the palaeo-
habitat in Nama Chu can be considered an independent archive of palaeo-precipitation, meltwater
events and other hydrological processes. The data from Nama Chu, however, can be scaled up for the
Bangong Co drainage basin and partly western Tibet because the general underlying palaeo-
hydrological processes were the same or at least very similar.

*1.3. Present climate conditions*




The climate is classified after Köppen-Geiger as cold desert, BWk (Peel et al., 2007). Meteorological
data are recorded at a station in Shiquanhe (also referred to as Ali, 32°30'N, 80°05'E, 4285 m a.s.l.),
ca. 110 km south of Nyak Co (Fig. 1). Limited data from an automated weather station, set up close to
the northern shore of Nyak Co, are in line with those from the station at Shiquanhe (Wen et al., 2016).
Precipitation is mainly westerly-derived (Zhang et al., 2011) and convective rainfalls occur (Fontes et
al., 1996; personal observation FR, 2012), which amount to 30-40% of the total rainfall (Maussion et
al., 2014). Nyak Co is located north but close to the normal northward extension of the SW Asian
summer monsoon (Gasse et al., 1991; Fontes et al., 1996; Wu et al., 2006; Tian et al., 2007). Wen et
al. (2016) reported a short-term monsoonal rainfall event of 25 mm with $\delta^{18}$O decreasing rapidly from
-9 to ca. -30‰, due to the amount effect in isotope fractionation (e.g. Kurita et al., 2009). The
weighted mean of $\delta^{18}$O in summer precipitation is -14.3‰, and is -18.8‰ in winter (Yao et al., 2013).
The $\delta^{18}$O in precipitation ranges from ca. -30 to -2.5‰ (Wen et al., 2016). Mean annual precipitation
is 70 mm (data from 1961 until 2009; Chinese Central Meteorological Office, 2010). Yu et al. (2007)
noted 75 mm, Yao et al. (2013) 82 mm. Inter-annual variation can be strong (Wen et al., 2016). The
annual potential evaporation can reach almost 2500 mm (Ou, 1981; Wen et al., 2016). The Bangong
Co drainage system is located in a permafrost region (Wang and French, 1995; Ran et al., 2015;
personal observations). The mean annual air temperature is 0.6°C (data from 1961 until 2009; Chinese
Central Meteorological Office, 2010). Minimum monthly temperatures of ca. -20°C occur in January,
maximum monthly temperatures are ca. 21°C during July (Ding et al., 2018). The lake is covered by
ice from November to April (Wang et al., 2014).

**2. Material and methods**

*2.1. Drainage basin studies and sample sites*

Fieldwork was conducted in September 2012. Geomorphology was mainly studied at Nyak Co, along
the northern Bangong Co shore as far as the third sub-basin west of Nyak Co, and in the Nama Chu





valley (Figs. 1-3). Observations included palaeo-shoreline, alluvial, periglacial and palaeo-glacial
features and the water flow direction in the chain of lakes. Electric conductivity, pH and water
temperature were measured. Water samples were taken for further analysis to the Freie Universität
Berlin. Sites and data of water samples not indicated in Fig. 2 can be found in Wilckens (2014). A
geological outcrop at the alluvial fan formed by Chiao Ho (Fig. 2; 33°37.629'N, 79°46.444'E, 4262 m
a.s.l. with GPS) exhibited fluvio-lacustrine sediments with well-preserved *Radix* and other shells. A
sediment sequence of 1.26 m thickness was sampled in 2 cm steps. The samples of ca. 200 g each
were packed in plastic bags and transferred to Freie Universität Berlin for further analyses. In the
Nama Chu valley, approximately 28 km east of Nyak Co, well-preserved fossil *Radix* shells were
collected from a few cm thick sediment sequence (Figs. 2 and 3; 33°32.018'N, 80°14.176'E, 4297 m
a.s.l. with GPS, 4286 m a.s.l. in SRTM) right below and in relation to a palaeo-shoreline. Additional
bulk sediment samples were taken for further analysis at the Freie Universität Berlin.

*2.2 Geomorphological maps, DEM and CORONA image*

The topography in Figs. 1 and 2 is based on 90 m elevation SRTM v4.1 data (Jarvis et al., 2008)
acquired year 2000. Catchment and sub-catchment boundaries and the drainage network were also
calculated based on the same data set using the Arc Hydro Tools package (ESRI, 2011) in ArcGIS
Desktop (ESRI, 2013) following standard workflows summarized in Dartiguenave (2007). Lake,
catchment and sub-catchment areas were calculated based on SRTM v4.1 data in a projected
coordinate reference system (WGS 84 / UTM zone 44N, EPSG: 32644) in ArcGIS Desktop (ESRI,
2013). Fig. 2 shows the extension and position of water bodies (incl. Nyak Co, dry and water filled
basins, and rivers) as of September 2012 according to two Landsat 7 imagery datasets (Entity IDs:
LE71460372012267PFS00 and LE71450372012260PFS00, acquired on 2012/09/23 and 2012/09/16,
respectively). The CORONA image used in Fig. 3 was purchased from the US Geological Survey
(Entity ID: DS1048-1134DA091; coordinates 33.480°N, 79.718°E; camera resolution: stereo medium;
acquisition date: 27-SEP-1968).




*2.3. Dating*

*2.3.1. Radiocarbon*

A *Radix* shell from the Nama Chu sediment sequence, two *Radix* shells from the Chiao Ho geological
outcrop and two charcoal samples from the same Chiao Ho sediment layers (Table 2) were dated at
Poznan Radiocarbon Laboratory. Fontes et al. (1996) calculated a lake reservoir effect for Nyak Co of
~6670 years. In the *Radix*-containing sediments from Nama Chu, we could not find charcoal particles
or other terrestrial organic remains for correcting the age but at Chiao Ho (Table 2). As Chiao Ho and
Nama Chu drain neighboring areas and the distribution of carbonatic rocks is similar in both sub-
catchments (Wang and Hu, 2004), we are confident that our age correction makes sense. The similar
but independent electron spin resonance (ESR) age supports this conclusion.

*2.3.2. Electron spin resonance (ESR)*

*Preparation and measurements*

*Radix* shell samples from the Nama Chu sediment sequence were gently crushed in a ceramic mortar
and sieved with 100 μm to remove finer material. Each aliquot containing 40 mg of the sample was
measured with an X-band JEOL FA-100 spectrometer. The measurement parameters used were 324 ±
5 mT magnetic field, 2mW microwave power, 0.1 mT modulation amplitude and scan time of 30s for
5 times. The single aliquot additive dose method was used to calculate the equivalent dose ($D_e$) using
$CO_2^-$ radical signal at g = 2.0006. The irridation of the sample was done with a Varian VF-50J X-ray
tube with tungsten target with 50 kV and 1 mA (Oppermann and Tsukamoto, 2015). Aliquots were
measured and irradiated within a thin quartz glass tube with 2 mm inner diameter. The sample tubes
were sealed with parafilm and were placed upside down during the X-ray irradiation (Tsukamoto et
al., 2015).




*Calibration of the X-ray dose*


The X-ray dose rate for calcium carbonate was calibrated using a modern coral sample. The coral

sample was crushed and sieved between 100-150 μm and divided into two sets. One set of the sample

coral was irradiated 75.6 Gy from a $^{60}$Co gamma source at the Technical University of Denmark. The

$CO_2^-$ signal (g =2.0006) from the coral from 3 aliquots of gamma irradiated coral (40 mg) were

measured with ESR using the same condition as the shell after preheating at 120°C for 2 minutes.

From the unirradiated set 3 aliquots were made and the same signal was measured after X-ray

irradiations for 60s and 120s and preheat at 120°C for 2 minutes. The ESR intensity of the gamma

irradiation coral was compared with the X-ray dose response curve. The gamma dose of 75.2 Gy is

equivalent to 90s X-ray irradiation (Fig. 4a). The X-ray dose rate for calcium carbonate was calculated

to $0.84 \pm 0.001$ Gy/s.

**Fig. 4**

*Equivalent dose measurements*

Four natural aliquots of the *Radix* shell were preheated between 100 and 130°C for 2 minutes with a

10°C increment (1 aliquot at each temperature) and the natural $CO_2^-$ radical signal was measured with

ESR. Then each aliquot was irradiated with 25 Gy X-ray, preheated at the same temperature and the

ESR signal intensity was measured again. This process was repeated 3 times. The $D_e$ values were

calculated by extrapolating the dose response curve to zero intensity (Fig. 4b). The $D_e$ values plotted

against the preheat temperature are shown in Fig. 4c. The $D_e$ values with preheats between 110 and

130°C are consistent with each other. Therefore, a preheat at 120°C was chosen and 3 more aliquots

were measured. The mean $D_e$ value from the 4 aliquots was calculated to $29.5 \pm 1.1$ Gy (Table 3).


*Dose rate and ESR age*




The external dose rate of the *Radix* shell was estimated using gamma spectrometry. About 5g sediment
sample surrounding the shells was sealed within a plastic cylinder about a month to ensure equilibrium
between $^{226}$Ra and $^{222}$Rn. The gamma rays from the sample were then measured using a Well-type
high resolution gamma spectrometer. The results are summarized in Table 3. The measured activity of
$^{238}$U is about twice as large as $^{226}$Ra. One possible explanation is that some $^{230}$Th has been lost from
the $^{238}$U decay chain (Long et al., 2014). Therefore, the external dose rate was divided into 2 parts, 1)
'supported part' which is originated from $^{232}$Th, $^{40}$K and the equilibrium part of $^{238}$U (calculated from
$^{226}$Ra activity) and 2) 'unsupported part' which is lost at $^{230}$Th (calculated based on $^{238}$U activity minus
$^{226}$Ra activity). The beta attenuation factors were calculated based on the thickness of the shell (70-80
μm) and the dose rate conversion factors of Guérin at al. (2011) were used. An alpha dose efficiency
of $0.1 \pm 0.05$ was assumed (e.g. Skinner, 1989). The cosmic dose rate was calculated following
Prescott and Hutton (1994).

**Table 3**

*2.4. Sediment processing and assignment and documentation of molluscan shells*

The sediment samples were washed and sieved using mesh sizes of 1, 0.5, 0.25 and 0.1 mm. The
sieved residue was visually analyzed using a Zeiss stereo microscope SV8. Shells of gastropods and
bivalves were picked for palaeo-environmental reconstruction and tiny pieces of charcoal were
separated for radiocarbon dating. Some shells were photographed using a Keyence VHX-1000
microscope (e.g. Fig. 5).

*2.5. Stable isotopes*

Five *Radix* shells from the Nama Chu site (Figs. 2 and 3) were selected for stable isotope analysis
based on shell preservation, completeness and sizes (Table 5; Fig. 5). *Radix* shells are built from



aragonite (e.g. Taft et al., 2012, 2013). First, the shells were cleaned in an ultrasonic bath and
subsequently any residual sediment particles were removed manually with a small brush. Sub-
sampling was conducted using a special dental drill device for milling the outer primary shell layer in
a constant distance along the ontogenetic order of the shell increments with a maximum depth of 50
μm. Up to 38 sub-samples were obtained from a single shell (Table 5), labeled in alphabetical order,
with [a] representing the ontogenetically latest shell part at the outer rim. The sub-samples of ~150 μg
were then measured for $\delta^{18}O$ and $\delta^{13}C$ ratios using a GasBench II linked to a MAT-253 ThermoFischer
Scientific™ isotope ratio mass spectrometer at Freie Universität Berlin. The measurements were
standardized against Carrara Marble (CAM) and Kaiserstuhl carbonatite in-house reference material
(KKS), which had been calibrated against Vienna PeeDee Belemnite (V-PDB) international isotope
reference material using NBS-18 and NBS-19. All results are reported in δ notation relative to V-PDB.
The external error (simple standard deviation) of the measurements is ± 0.06‰ for $\delta^{18}O$ and ± 0.04‰
for $\delta^{13}C$.

**3. Results**

*3.1. Dating*

*3.1.1. Radiocarbon*

A *Radix* shell sampled from the Nama Chu sediment layer was dated to 12670 ± 60 years B.P. Two
*Radix* shells from the Chiao Ho alluvial section were dated to 8540 ± 40 and 8480 ± 40 years B.P.
Two charcoal samples from the same Chiao Ho sediment layers were dated to 2275 ± 30 and 2400 ±30
years B.P. The [14]C ages of the two charcoal samples were subtracted from the ages of the two Chiao
Ho *Radix* shells. Consequently, the lake reservoir effects for the latter shells are 6265 and 6080 [14]C
years B.P. A mean lake reservoir effect of 6172 [14]C years B.P. was then subtracted from the age of the
Nama Chu shell which results in 6498 years B.P. Calibration of 6498 years B.P. with CALIB (Stuiver
et al., 2013, online executive version 7.0html) resulted in a weighted average age at 2σ precision of



7393 ± 114 cal. years B.P. The radiocarbon age of the Nama Chu sediment horizon is therefore
considered ~7.4 ± 0.1 cal. ka B.P., which is early Middle Holocene. The data are compiled in Table 2.

**Table 2**

*3.1.2. Electron spin resonance*

The internal U content of the fossil *Radix* sample from Nama Chu was not measured. The U content of
modern *Radix* shells from Bangong Co was measured and the mean value is 0.05 ± 0.01 (n = 6;
Wassermann, 2014). However, it is well known that shells take up U from surroundings (e.g. Grün,
1989). Schellmann et al. (2008) reported a mean U content of Holocene shells (> 2.5 ka) to be 2.8 ±
2.7 ppm (n = 63). Assuming this mean U content as the current U content of the shells, we calculated
the age by two scenarios: U content increased linearly with time (linear uptake, LU) or the uptake
occurred at an early stage of the burial time (early uptake, EU). The calculated ages are 8.1 ± 1.0 ka
(LU) and 7.4 ± 1 ka (EU) respectively (Table 3).

**Table 3**

*3.2. Features of the early Middle Holocene habitat inferred molluscan shells*

The sandy to fine-gravelly deposits sampled at a Nama Chu pond palaeo-shoreline (Fig. 3) exhibited
shells from four molluscan genera (Fig. 3C). Shells of the aquatic gastropods *Radix* sp. and *Gyraulus*
sp. were fairly abundant. In comparison, shells of the bivalve *Pisidium* sp. occurred less frequently and
from the gastropod *Valvata* sp. only single shells were found. The ecological traits of these genera,
which provide information about the palaeoenvironment, are compiled in Table 4.

**Table 4**



*3.3. Shell morphology and δ¹⁸O and δ¹³C values in early Middle Holocene* Radix

Prior to sub-sampling (micro-milling), the selected five shells, termed NC1-5, were measured in height
and the number of whorls were counted. These data and the individual number of sub-samples are
compiled in Table 5. As an example, the shell NC2 is figured (Fig. 5).

**Fig. 5**

**Table 5**

All sclerochronological isotope patterns and single isotope values are shown in Fig. 6 and Table 6,
respectively. The range of $\delta^{18}O$ values in all five shells that were analyzed, is from -10.2‰ in shell
NC3 to -2.5‰ in shell NC5. The mean oxygen isotope compositions of shells NC1-4 are in the range
between -9.2 and -7.5‰. Shell NC5 exhibits a mean $\delta^{18}O$ value of -4.6‰. The range of $\delta^{13}C$ values in
all five shells analyzed is from 3.2‰ in NC4 to 8.4‰ in NC1. The mean carbon isotope values of the
shells are in the range of 4.9 to 6.5‰. The correlations between oxygen and carbon stable isotope
patterns are $r^2$= 0.8 for NC1, 0.5 for NC2 and NC3, 0.4 for NC4 and 0.8 for NC5.

**Fig. 6**

**Table 6**

**4. Discussion**

*4.1. Age of* Radix *shells*

Two dating methods were applied; radiocarbon, which produced an age for the Nama Chu shells of
~7.4 ± 0.1 cal. ka B.P., and electron spin resonance, which gave an age of 8.1 ± 1 ka (LU model) and



7.4 ± 1 ka (EU model). The inferred lake reservoir effect of ~6200 years suggests strong detrial input
of old carbon. Fontes et al. (1996) calculated a lake reservoir effect of 6670 years for a sediment core
taken from central eastern Nyak Co. On the one hand this difference of ~500 years does not
significantly increase temporal uncertainty of our early Middle Holocene case study and on the other
hand it could be due to higher detrial input of Jurassic limestone by the Makha River (Fig. 2), which is
draining the southern catchment of Nyak Co. The EU-model-ESR age of 7.4 ± 1 ka is very similar to
the radiocarbon age of 7.4 ± 0.1 ka. We thus tentatively consider an approximate age of ~7.5 ka to
address the palaeo-habitat and are confident to report early Middle Holocene processes. Although the
five shells used for stable isotope analyses came from a single few cm thick sediment layer, we
assume that they rather reflect a multi-decadal period, and thus represent environmental archives of
five different years. This assumption is supported by the sclerochronological stable isotope patterns
(Table 6; 4.3.).

*4.2. Habitat simulation with the aid of early Middle Holocene aquatic molluscs*

The fossil assemblage indicates a shallow littoral environment (Table 4). This is in line with the
observation that the sediments were deposited along a palaeo-shore. We therefore use the palaeo-
shoreline as contemporary water level (Fig. 2C). Considering the reconstruction of Dortch et al. (2011)
that the Early to Middle Holocene lake level of Nyak Co was ca. 10 m higher than nowadays, the
difference to the level of the Nama Chu pond was ca. 35 m. When the pond was filled up to the
palaeo-shoreline, it was approximately 5-6 m deep and interconnected with the neighboring ponds
(Fig. 2C). The short-term grain size changes, e.g. from fine sand to fine gravel, show that there were
significant hydrological changes, but generally it can be assumed that it was a lacustrine to semi-
lacustrine habitat. Salinity was in the range of freshwater to oligohaline (Table 4).

*4.3. $\delta^{18}O$ and $\delta^{13}C$ values in shells from early Middle Holocene* Radix *shells*

*4.3.1. Mean values and range of values*




The range of mean $\delta^{18}O$ values of the five shells from -9.2 to -4.6‰ indicates that the Nama Chu
palaeo-habitat was located in a dynamic hydrological system. This becomes even more evident when
comparing the most negative (-10.2‰) and the least negative (-2.5‰) values. Modern shells from
Nyak Co have mean values of -2.18 and -2.23‰ (Taft et al., 2013). Several authors who studied
precipitation or aquatic systems on the plateau (e.g. Fontes et al., 1996; Cai et al., 2012; Wünnemann
et al., 2018) have argued that not temperature but precipitation source and amount, and evaporation are
the dominant factors in oxygen isotope fractionation. Based on precipitation recorded at Shiquanhe
(Fig. 1), Yu et al. (2007), however, concluded that variations in $\delta^{18}O$ relate closely to temperature
variations. Bangong Co water, on the other hand, was considered to be mainly controlled by local
relative humidity (Wen et al., 2016). We interpret the lower $\delta^{18}O$ values of Nama Chu compared to
Nyak Co primarily as an effect of shorter water residence time, i.e. the water is less influenced by
evaporation. Mean -9.2‰ indicates a significantly stronger water flow than mean -4.6‰. The range
reflects semi-lacustrine to lacustrine conditions. A correlation of $r^2 = 0.8$ between oxygen and carbon
stable isotope values (Table 6) in shells NC1 (mean $\delta^{18}O$ of -7.5‰) and NC5 (mean $\delta^{18}O$ of -4.6‰)
indicates some degree of covariance, which is typical for closed-basin lakes and ponds (Li and Ku,
1997; Taft et al., 2013). The closed-basin periods, however, likely did not last for long because the
freshwater molluscan assemblage demonstrates that salinity did not vary much. On the other hand,
modern shells from Nyak Co show rather low $\delta^{18}O$, which co-varies with $\delta^{13}C$ (Taft et al., 2013),
indicating a closed basin but the lake spills over to its neighboring basin and salinity is low. The water
sources of the Nama Chu palaeo-habitat are discussed under 4.3.2.

Mean $\delta^{13}C$ values of the five shells are in a range of 4.9 to 6.5‰, which is exceptionally

positive, compared to other (semi-) lacustrine systems (e.g. Leng and Marshall 2004) but in line with
$\delta^{13}C_{DIC}$ from modern sediments of the Nama Chu pond (Table 1). Shells of *Radix* sp. living in Nyak
Co show mean $\delta^{13}C$ values of -2.35 and -2.48‰ (Taft et al., 2013). Fontes et al. (1996), however,
reported $\delta^{13}C$ values from early Holocene Nyak Co carbonates of up to 7.2‰, which they related to
enhanced aquatic photosynthesis during evaporative shallow lake conditions and/or to some methane
formation within bottom sediments. Other regional high $\delta^{13}C$ values were found in Sumxi Co (Fig. 1)



carbonates (Fontes et al., 1993) and in a Tso Moriri (Fig. 1) sediment core (Mishra et al., 2015). The
latter authors concluded that the role of phytoplankton productivity was minimal because of
oligotrophic conditions (Mishra et al., 2015). Goto et al. (2003) reported similar high $\delta^{13}C$ values from
central Tibetan Plateau Siling Co, which they related to evaporation (Stiller et al., 1985). The even
more positive $\delta^{13}C$ values of carbonates from Lake Caohai (China, Guizhou Province) were explained
by bacterial degradation of aquatic organic matter, generating methane, preferentially $^{12}CH_4$, and
leading to an enrichment of $^{13}C$ in the lake water and carbonate (Zhu et al., 2013).

We consider that the high $\delta^{13}C$ values were likely triggered by a combination of the cited

factors plus detrital input. Effective evaporation is reflected by corresponding $\delta^{13}C$ and $\delta^{18}O$ values
(Horton et al., 2016). The most negative $\delta^{18}O$ value is from the same shell as the least positive $\delta^{13}C$
value; the least negative $\delta^{18}O$ value is from the same shell as the most positive $\delta^{13}C$ value; etc. Organic
productivity was probably higher than in Nyak Co, due to the fact that the Nama Chu pond was only
5-6 m deep, and light could penetrate to the bottom and trigger photosynthetic processes in all water
layers. Aquatic plants and algae utilize $CO_2$ as source of carbon for photosynthetic processes with
preferable uptake of $^{12}C$ (Chikaraishi, 2014), which leads to a $^{13}C$ enrichment. Methane bubbling was
observed (2009, 2012) and it is likely that organic-rich mud was available for microbes also during the
early Middle Holocene. Liu et al. (2017) outlined methanogenic pathways from a short Nyak Co
sediment core. $\delta^{13}C_{CH4}$ was in a range of ca. -60 to -110‰. Seasonal permafrost thawing could have
triggered another methanogenic pathway (Rivkina et al., 2007). Permafrost represents a considerable
carbon pool (Wagner et al., 2007; Mackelprang et al., 2011). Methane bubbling means that preferably
$^{12}C$ was removed from the Nama Chu palaeo-habitat, leaving the remaining carbon $^{13}C$ enriched
(Walter et al., 2006). Detrital input of old carbon from Jurassic and Permian limestone (Wang and Hu,
2004) is considered to represent another cause for the high $\delta^{13}C$ values. The limestones probably
represent shallow water tropical carbonate formations which may exhibit $\delta^{13}C$ values as positive (e.g.
Isozaki et al., 2007) as was measured in the fossil *Radix* shells. The relatively high $^{14}C$ reservoir effect
in the Bangong Co system (Fontes et al, 1996; this study) indicates detrital input of old carbon.

*4.3.2. Sclerochronological patterns*




The palaeo-environmental setting suggests that the Nama Chu pond was sensitive to short-term
atmospherical, hydrological, limnological and hydromorphological changes. It thus can be expected
that the five *Radix* shells, which were formed in equilibrium with the pond water, archive early Middle
Holocene hydrological signals over their life spans of ca. 12-15 months (Taft et al., 2012, 2013). The
sediment sequence from which the shells were sampled represents a multi-decadal period, and the
individual ranges of stable isotopes and their mean values (4.3.1.) indicate that the five shells reflect
five different years around ~7.5 ka. The interpretation of isotopic signatures of the shells is based on
the following considerations:

a) Precipitation source: Regional monsoonal (summer) precipitation has mean $\delta^{18}O$ values of

ca. -14 to -16‰ (Yu et al., 2007; Yao et al., 2013) and can be as low as -30‰ in case of short-term
heavy rainfall (Wen et al., 2016). Monsoonal rainfall is therefore isotopically lighter than the pond
water and negative excursions can be expected in the isotope patterns of the shells. Convective clouds
form by regional moisture evaporation particularly during May to October when the lake surfaces are
not covered by ice. Seasonal permafrost thawing provides soil moisture, which becomes part of the
convective system. Potential monsoonal rainfall would add to the soil moisture. Measured $\delta^{18}O$ values
of regional convective rainfall range from ca. -5.5 to 0.2‰ (Fontes et al., 1996; Mishra et al., 2014)
and thus at least in shells NC1-4 positive isotopic excursions can be expected, in case of a significant
amount. June snowfall over Tso Moriri (Fig. 1) exhibited a $\delta^{18}O$ value of -22.4‰ (Biggs et al., 2015).
Regional snowfall accumulates mainly in winter and is westerly-derived (Biggs et al., 2015). The
oxygen isotope composition of local rivers is dominated by meltwater and ranges from ca. -12 to -
14‰ in the Nyak Co catchment (Fontes et al., 1996; Wilckens, 2014; Wen et al., 2016). Meltwater
pulses thus will lead to negative excursions in $\delta^{18}O$ patterns of the Nama Chu shells. Regional
meltwater increases in May and peaks in July (personal communication with local people, 2012). In
the case of Nama Chu snowmelt and permafrost thawing have to be considered.

b) Precipitation amount: Quantification is difficult but rainfall, which is intensive enough to

wash in soil, can be identified using $\delta^{13}C$ (Taft et al., 2012, 2013). $\delta^{13}C$ of dissolved soil carbonate
from Ladakh revealed values from ca. -20 to -28‰ (Longbottom et al., 2014). Dissolved organic





531 carbon from terrestrial plants is in a similar range (Cloern et al., 2002; Wynn et al., 2007). The mean

532 $\delta^{13}C$ values of the Nama Chu shells range from 4.9 to 6.5‰. Carbon washed in from soil thus would

533 lead to negative isotope excursions in the sclerochronological patterns.

534  c) Evaporation and ice cover period: From November to April, Nyak Co is covered by ice

535 (Wang et al., 2014), and it can be assumed that the surface of the Nama Chu palaeo-pond was frozen

536 for a similar period although possibly with a seasonal lag of some weeks due to the lower water depth.

537 The ice cover period may have been shorter during the early Middle Holocene. Ice cover prevents

538 exchange between atmosphere and pond water, and potential changes in isotope composition must be

539 intrinsic. Consequently, variation of oxygen isotope values is considered to be low during ice cover

540 conditions, carbon isotope values, however, decrease due to reduced productivity under lower light

541 penetration and lower temperatures. Evaporation is effective from approximately May to October and

542 leads to heavier isotope values but is potentially superimposed by meltwater inflow and rainfall. The

543 effect of evaporation can be seen best regarding the dry period after summer rainfall until the

544 beginning of ice cover (Taft et al., 2013). The mean isotope values of the shells (Table 6), however,

545 show clearly the inter-annually varying influence of evaporation.

546  d) Organic productivity:  Main controlling factors are light and temperature (Chikaraishi,

547 2014) and thus periodically higher $\delta^{13}C$ shell values reflect the summer season while lower $\delta^{13}C$ shell

548 values indicate reduced productivity of water plants and algae during winter. The productivity of

549 microbes is exemplified by archaeans and briefly outlined in the next paragraph.

550  e) Methanogenesis: During biogenic methane production preferably $^{12}C$ is processed (Walter

551 et al., 2006). While the gas will leave the habitat by bubbling during the summer months triggering $^{13}C$

552 enrichment in the water, it may accumulate under ice in winter. It was observed (FR, 2013) on the

553 eastern Tibetan Plateau that *Radix* moves on the underside of pond ice and likely consumes algal

554 growth there. Thus it can be expected that methanogenesis is occasionally archived in *Radix* shells.

555 Recent observation (2009, 2012) of methane bubbling in the Nama Chu pond hints at this possibility.

556  f) Temporal resolution: Although *Radix* is active in all seasons and even under ice, it grows

557 much slower in winter than in summer (Gaten, 1986). Data from modern *Radix* shells indicate that

558 growth was ca. three times slower during the ice cover period compared to the average ice-free period



(Taft et al., 2013). The temporal resolution of the summer isotopic signals archived in the shell is thus
significantly higher (ca. weekly) than of those archived in winter.

It is unlikely that the five early Middle Holocene *Radix* individuals hatched and died during the same
time of a year but represent records of different length and seasonality. Maximum shell height and
number of whorls are notably lower in NC5 (Table 5) suggesting that this individual had a
comparatively shorter lifespan. Ice cover periods identified in the isotope patterns can, for example, be
used to infer the chronology of the individual isotope patterns. The ice cover period in shell NC2 (Fig.
6) is from [t] to [p]. $\delta^{18}O$ shows little variation in this shell section and $\delta^{13}C$ values are relatively low.
Interestingly $\delta^{13}C$ increases temporarily around [s]. We speculate that methanogenesis was responsible
for this effect. A similar $\delta^{13}C$ excursion can be seen in shell NC1 (Fig. 6) where the ice cover period is
from [p] to [k] and is even more significant (double peak) in shell NC3 where the ice cover period is
from [q] to [l]. In shell NC4, $\delta^{18}O$ shows little variation from [t] to [a], which is much too long a
period for ice cover, suggesting superimposition by other factors. The lowest $\delta^{13}C$ values imply that
ice cover was roughly from [s] to [n]. In shell NC5, we tentatively appoint the ice cover period to [o]
to [k]. $\delta^{18}O$ shows little variation and $\delta^{13}C$ values are low here. Using this seasonal marker, we discuss
the complete isotope patterns of NC1-5 in ontogenetic chronology (Fig. 6).

Shell NC1: This gastropod hatched during early summer. The general trend of $\delta^{13}C$ shows

increasing productivity to [w], which is overprinted by two negative excursions, with minimum values
at [z10] and [z2]. During the first negative $\delta^{13}C$ excursion, $\delta^{18}O$ decreases correspondingly. We
interpret this as significant monsoonal precipitation, bringing isotopic lighter rain into the pond and
triggering the inwash of isotopically light soil carbon. The second negative $\delta^{13}C$ excursion is
accompanied by a positive $\delta^{18}O$ peak at [z3]. Again a significant inwash of soil carbon occurred but
this time triggered by convective rainfall. The following increase of $\delta^{18}O$ to [x] is considered to reflect
the dominance of evaporation, which is in line with the increasing $\delta^{13}C$ values. The subsequent shift to
lighter stable isotope values represents the transition to winter conditions, with reduced evaporation
and decreasing primary bioproductivity. The ice cover period from [p] to [k] is followed by an
increase of bioproductivity until again summer conditions were reached. $\delta^{18}O$ was strongly dominated





by evaporation suggesting that there was little snowfall during the preceding winter and thus no
significant influence by meltwater. The gastropod died before potential (second) summer rainfall
events occurred. Occasional light rain could have fallen but cannot be detected in the isotope pattern
because of signal weakness.

Shell NC2: This gastropod hatched when significant monsoon moisture penetrated the western

Tibetan Plateau, likely during middle summer. Indication are negative excursions of $\delta^{18}O$ [z7] and
$\delta^{13}C$ [z6]. Inwash of light terrestrial carbon stopped immediately with the termination of heavy rainfall
leading to a steep increase of $\delta^{13}C$ to the high summer bioproductivity level [z3]. The subsequent
increase of $\delta^{18}O$ to [z3] is due to evaporation dominating potential lighter rainfalls and snowmelt. A
second monsoonal rainfall period is indicated by abrupt decreases at [z2] of $\delta^{13}C$ and $\delta^{18}O$. The
following steep increase of $\delta^{18}O$ is due to evaporation and likely represents September, when rainfall
amounts were low and meltwater played a minor role, due to lower temperatures. Such a September
pattern was found in modern *Radix* sp. from Nyak Co (Taft et al., 2013). The subsequent turnover of
isotope signatures to lighter values can be explained by increasingly weaker insolation and colder
temperatures, likely during October ~7.5 ka. In November the pond became ice covered ([t] to [p]).
The following spring (May) primary bioproductivity increased quickly and the pattern shows no
negative excursions until [b]. The simultaneous increase of $\delta^{18}O$ is modest to [e] but stronger
afterwards to [b], likely because the evaporation signal could dominate the meltwater signal only in
summer. The synchronous abrupt negative excursions in $\delta^{13}C$ and $\delta^{18}O$ from [b] to [a] may indicate a
monsoonal moisture pulse.

Shell NC3: This *Radix* individual hatched in early summer when primary bioproductivity

started to increase significantly. A small negative excursion of $\delta^{13}C$ and an even smaller negative peak
of $\delta^{18}O$ at [z2] may represent a monsoonal moisture pulse. With a mean of -9.2‰, $\delta^{18}O$ of the pond
was relatively negative and thus monsoonal rainfall is likely not well evidenced in the pattern. The
negative $\delta^{18}O$ peak at [x] is considered a meltwater pulse and not heavy rainfall because $\delta^{13}C$ did not
react. $\delta^{13}C$ and $\delta^{18}O$ peak at [u], likely in September. Autumn turnover is indicated by steep decreases
of $\delta^{13}C$ and $\delta^{18}O$ towards the ice cover period [q] to [l]. Spring (May) is characterized by increasing



$\delta^{13}C$ and $\delta^{18}O$ values to [e]. A simultaneous drop of both isotope values to [c] may exhibit a
monsoonal moisture pulse.

Shell NC4: This snail hatched in late summer because primary bioproductivity was already

quite high. The period to [z] may represent September because of evaporation dominating $\delta^{18}O$. In
September ~7.5 ka, just before the autumn turnover started, strong convective rainfall is evidenced by
$\delta^{18}O$ becoming heavier and the abrupt negative excursion of $\delta^{13}C$. Autumn (October) turnover is
clearly indicated by decreasing stable isotope values towards the ice cover period which was likely
from [s] to [n]. The following spring (May) is characterized by increasing bioproductivity, $\delta^{18}O$,
however, showing a negative trend. This may be explained by meltwater dominating evaporation. It is
possible but unlikely that the negative excursion of $\delta^{13}C$ at [b] was caused by rainfall because $\delta^{18}O$ of
the pond water remained unchanged at ca. -8‰, and rainfall with similar values is difficult to infer for
the region as no such value has been reported.

Shell NC5: This individual hatched in late summer when evaporation became dominant and

primary bioproductivity reached its maximum. The simultaneous negative peaks of $\delta^{13}C$ and $\delta^{18}O$ at
[r] indicate a September monsoonal moisture pulse. September is inferred because of the evaporation
signal still increasing due to rainfall/humidity ceasing. On the other hand, bioproductivity had started
to decrease. The autumn turnover terminated with the beginning of the ice cover period which is
approximately from [o] to [k]. The following spring (May) triggered bioproductivity ($\delta^{13}C$) and
evaporation began dominating the $\delta^{18}O$ values to [d]. Subsequently, both isotope values drop, which
we consider the pattern of monsoonal moisture penetration into the area (Fig. 6).

Four out of five of the sclerochronological stable isotope records exhibit significant rainfall periods. In
shell NC3 the signals are less clear, which might be due to the generally lighter $\delta^{18}O$ of the palaeo-
pond water. Five to eight rainfall events are related to monsoonal moisture, while two events evidence
strong convective rainfall. Shell NC2 indicates that two monsoonal moisture pulses could appear
during one season. Isotope patterns of modern *Radix* shells from lake basins with regular monsoonal
precipitation, Bangda Co and Donggi Cona (eastern Tibetan Plateau), reveal single extended events,
relating to the summer rain season (Taft et al., 2012, 2013). The monsoonal behavior in the study area



on the western Tibetan Plateau was thus quite different. The data suggest that during the early Middle
Holocene the monsoonal moisture did not penetrate much further onto the plateau than nowadays,
with the difference, rainfall events/periods happened more regularly and were stronger. Two shell
patterns (NC1, NC4) indicate convective rainfall, which we consider stronger or more extended than
those observed in modern times. This can be explained by higher summer insolation (Berger and
Loutre, 1991), moister soils due to monsoonal precipitation and much more extended lake surfaces
(Liu et al., 2013) around ~7.5 ka. The average annual precipitation amount was likely several times
higher than nowadays. These suppositions are basically in line with other early Middle Holocene
records from the western plateau (Gasse et al., 1991, 1996; Fontes et al., 1993; Brown et al., 2003;
Wünnemann et al., 2010).
There are no glaciers located in the Nama Chu catchment and there is no indication that it was
different under early Middle Holocene climate. The amount of meltwater that reached the palaeo-pond
was therefore mainly dependent on westerly-derived snowfall during winter. Summer snowfall occurs
nowadays but snow normally melts within hours to a couple of days (personal observations) and thus
does not add to spring meltwater from accumulated winter snowfall. These processes unlikely changed
during the Holocene. The role of seasonal permafrost thawing for the hydrological system remains
unclear. The five shell patterns indicate inter-annual differences in meltwater amounts. While in shells
NC1 and NC5 little influence of meltwater on $\delta^{18}O$ can be inferred, the influence is significant in shell
NC2, the isotopically light meltwater mitigating the evaporation signal during spring and early
summer. The strong meltwater pulse identified in NC3 can be possibly related to the outburst of a
meltwater-fed pond in the upper catchment of Nama Chu. The domination of meltwater in the isotope
pattern of NC4 is in line with the pattern of a modern *Radix* from southern Nyak Co (Taft et al., 2013),
sampled not far from the mouth of the Makha River (Fig. 2), which is draining meltwater (personal
observation FR, 2012). Based on our data, we suggest that the westerly influence during ~7.5 ka
winters was similar to modern times.
The ice cover period during ~7.5 ka was recorded by all shells. The data, however, do not
allow to infer whether the length of the ice cover period differed from the modern situation. This is



also due to the weak observational record for comparison and that we do not have a good modern
analogue for the Nama Chu palaeo-pond.

The influence of biogenic methane production on $\delta^{13}C$ is likely, due to the high mean values in

the context of methane bubbling observation. On the other hand, specific positive excursions during
the ice cover period, cannot easily be explained by primary producers' productivity pulses. We
consider the influence of methane production on the $\delta^{13}C$ of certain ponds or lakes underestimated.

**5. Conclusions**

The sclerochronological isotope patterns of early Middle Holocene *Radix* shells are suitable to report
hydrological processes from the western Tibetan Plateau in ca. weekly (summer) to sub-monthly
(winter) resolution over the lifespan of the gastropod, which is about one year.

We infer from our data that i) monsoonal rainfall reached the area more regularly and in

higher amounts than nowadays; ii) monsoonal rainfall did not prevail over the whole summer season
but penetrated the western Tibetan Plateau as extended moisture pulses; iii) the northern boundary of
the SW Asian summer monsoon was in a similar position as in modern times but the monsoonal
system was more dynamic; iv) significant convective rainfall occurred and can be clearly distinguished
from monsoonal precipitation with the aid of stable isotope patterns; significant convective rainfall is
due to higher summer insolation (evaporation), higher soil moisture (by monsoonal penetration) and
much larger lake surface areas during ~7.5 ka; v) isotopic signals of monsoonal and regional
convective precipitation can be clearly differentiated from meltwater signals in the records; vi) in the
study area, the meltwater amount correlates with westerly-derived winter snowfall amount; the
snowfall amount during the early Middle Holocene was probably similar to modern times; vii)
biogenic methane production could likely be identified in the isotope patterns and is possibly
underestimated in lake systems.

**Author contribution**



LT and FR prepared the original manuscript with contributions from all co-authors. FR conceptualized
the overarching research goals and aims. ST developed the design of the dating methods. LT and UW
performed and interpreted the stable isotope data. HC was responsible for the coordination of the
research activity planning and execution. CA and TW investigated the ecological traits of the
molluscs. CL was responsible for the visualization and presentation of the data.

**Acknowledgements**
Catharina Clewing (Giessen University, Germany) and Marc Weynell (FU Berlin) greatly assisted
Frank Riedel during fieldwork on the western Tibetan Plateau in 2012. We appreciate that Maike Glos
(FU Berlin) processed the sediments. She also micro-milled the gastropod shells and prepared the
samples for stable isotope analyses. Many thanks to Atsushi Suzuki (Geological Survey of Japan) and
Mayuri Inoue (Okayama University) who provided the modern coral sample for the X-ray calibration
for ESR dating. The gamma irradiation was made with the help of Jakob Helt-Hansen, Jim Thorslund
Andersen and Kristina Thomsen (all Technical University of Denmark). Thanks to Tomasz Goslar
(Poznan, Poland) for determination of the radiocarbon ages. Jan Evers (FU Berlin) kindly improved
figures. We are grateful to the German Science Foundation (DFG) for financial support. This is a
contribution to the DFG priority program TiP.

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

**Figure captions**





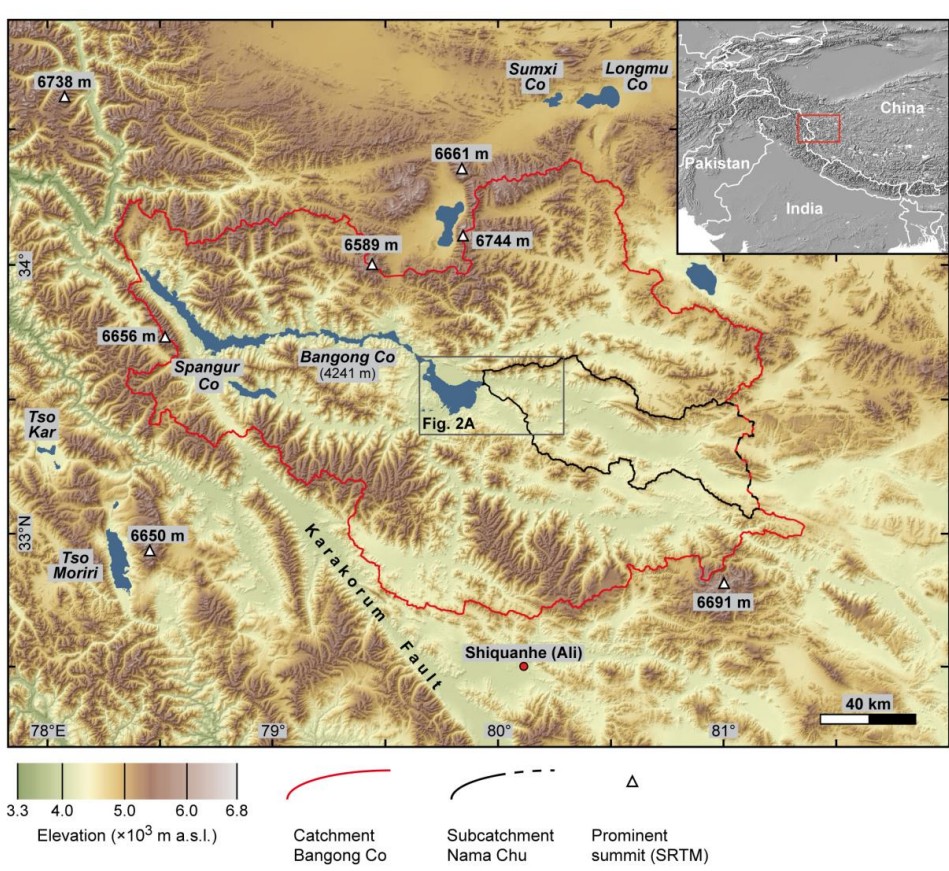


Fig. 1. Digital elevation model (SRTM) of part of the western Tibetan Plateau showing the

transboundary Bangong Co drainage basin with lake system, total catchment (red line) and sub-

catchment of Nama Chu valley (black line). All elevations are derived from SRTM.



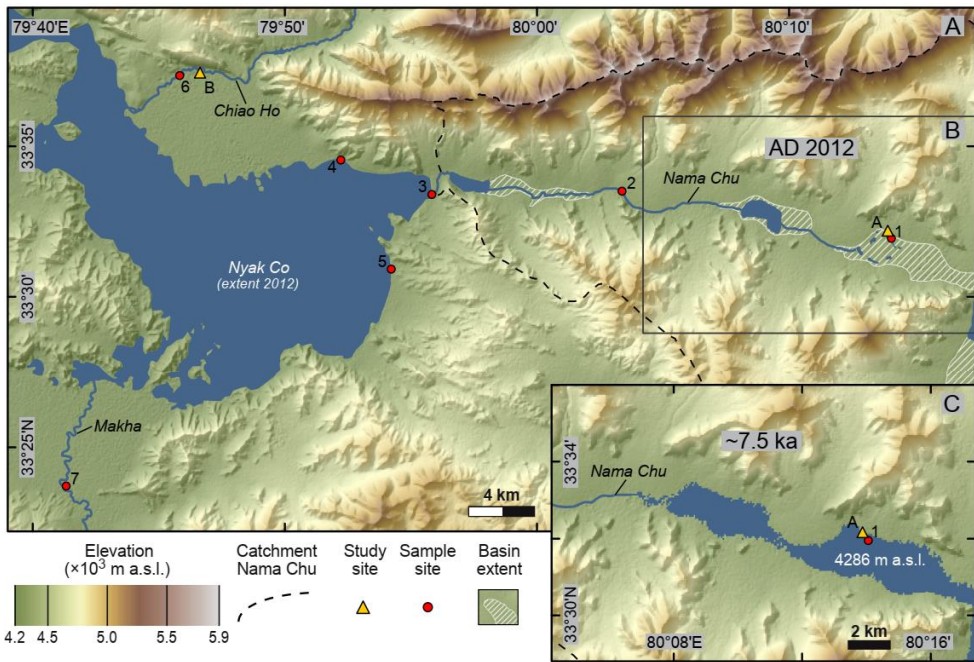


**Fig. 2. A.** Digital elevation model (SRTM) of the eastern Bangong Co system (Nyak Co) with

locations of water samples, study sites and major tributaries indicated by symbols. **B.** Focal area as in

2012. **C.** Water extension of focal area simulated for ~7.5 ka, based on the morphology of basins,

palaeo-shorelines and the altitudinal position of fossil shell bearing fluvio-lacustrine sediments.



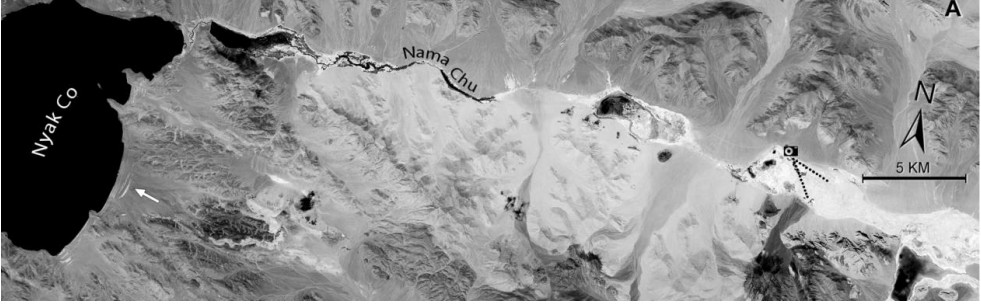

1055

**Fig. 3. A.** CORONA satellite image of western Nama Chu valley and Nyak Co, the easternmost lake

basin of the Bangong Co system (compare Fig. 1); arrow: palaeo-shoreline features. Camera symbol

and dotted lines refer to figure 3B; **B.** Photograph taken in September 2012 showing the modern

setting of the studied palaeo-hydrological system in Nama Chu valley; the greyish undulated

landscape between the grassland and the mountains represents frozen mounds of lacustrine sediments,

formed by permafrost processes; **C.** Littoral sediments of ~7.5 ka age, from which the studied

molluscan shells were sampled (Handheld GPS for scale); the sediments were found along a palaeo-

shoreline.




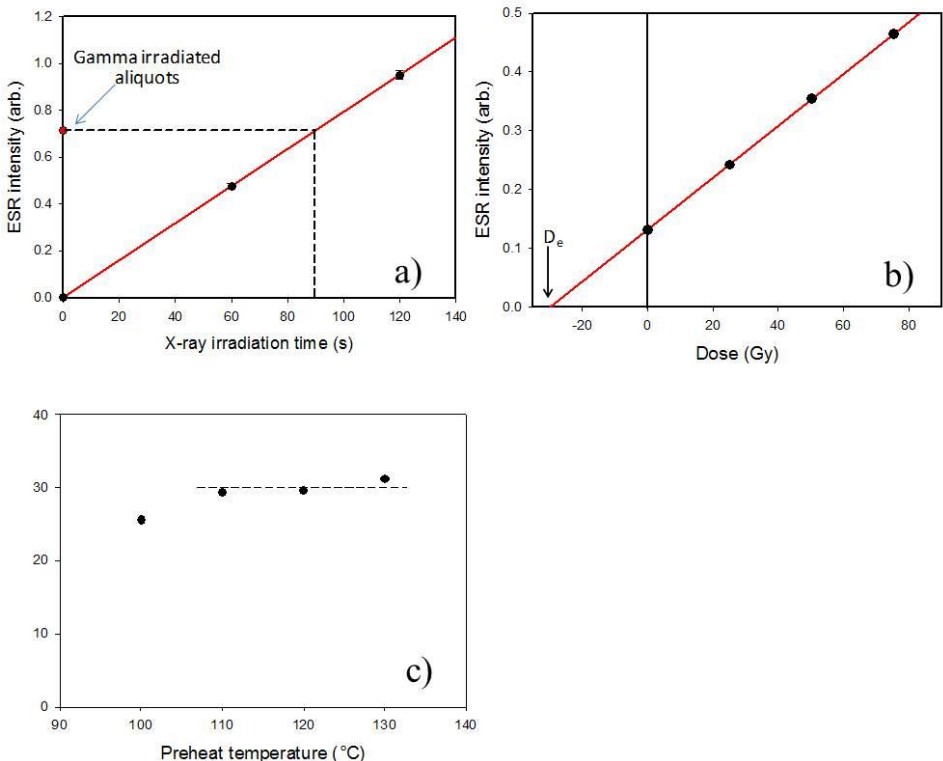


**Fig. 4. A.** X-ray calibration of calcium carbonate using a modern coral sample. Each data point is the
mean of 3 aliquots. **B.** Single aliquot additive dose $D_e$ measured from one aliquot of *Radix* shell
preheated at 120°C. **C.** $D_e$ values of the *Radix* shell sample measured at different preheat temperatures.

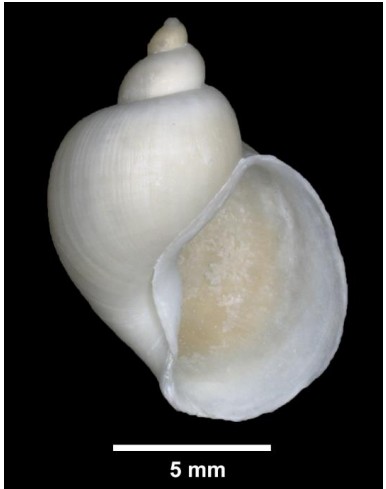




**Fig. 5.** One (NC2) of the five (NC1-5) fossil *Radix* sp. shells from Nama Chu valley, which were sub-
sampled sclerochronologically for stable isotope analyses.

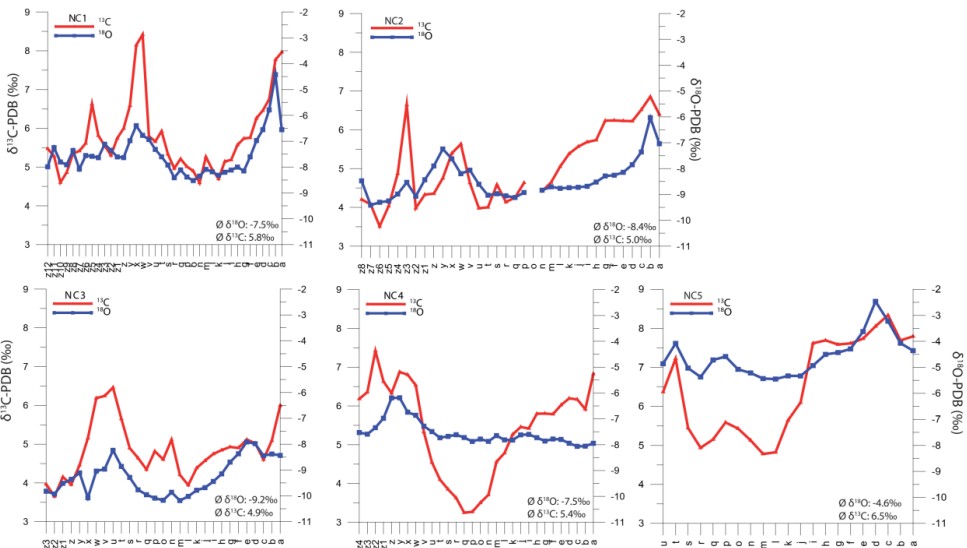


**Fig. 6.** Sclerochronological δ¹⁸O and δ¹³C patterns from studied early Middle Holocene *Radix* sp.
shells (NC1-NC5) sampled from Nama Chu valley sediments.

**Table captions**

**Table 1.** Selected water parameters from Nyak Co (3-5) and two tributaries (6 and 7) and from Nama
Chu (1 and 2). Except for electric conductivity, psu, T and pH, all analytical data from Wilckens
(2014). Location numbers refer to Fig. 2.

| Map location | 1 | 2 | 3 | 4 | 5 | 6 | 7 |
|---|---|---|---|---|---|---|---|
| Geographical | 33.5326°N 33.3945°N | 33.5581°N | 33.5573°N | 33.5767°N | 33.5152°N | 33.6219°N | |
| coordinates | 80.2346°E 79.6880°E | 80.0558°E | 79.9319°E | 79.8695°E | 79.9043°E | 79.7633°E | |
| Sampling date | 16.09.2012 17.09.2012 | 16.09.2012 | 11.09.2012 | 14.09.2012 | 11.09.2012 | 14.09.2012 | |
| Water T in °C | 20.2 | 23.3 | 13.8 | 19.2 | 18.2 | 13.3 | 11.2 |



| | | | | | | | |
|---|---|---|---|---|---|---|---|
| EC µS/cm | 41600 | 731 | 786 | 909 | 920 | 555 | 239 |
| psu | 29.89 | 0.37 | 0.51 | 0.51 | 0.53 | 0.35 | 0.16 |
| pH | 8.8 | 8.4 | 9.3 | 9.3 | 8.9 | 8.8 | 8.5 |
| $\delta^{18}O_W$ ‰ | n.a. | -13.0 | -8.4 | -3.9 | -3.5 | -13.9 | -13.3 |
| $\delta D_W$ ‰ | n.a. | -91.5 | -75.7 | -48.8 | -48.6 | -100.2 | -99.3 |
| $\delta^{13}C_{DIC}$ ‰ | +6.2 | +0.7 | +0.1 | +2.4 | +2.2 | -1.0 | -6.5 |
| $HCO_3$ µmol/l | 22786 | 4827 | 3327 | 4338 | 5390 | 3686 | 1875 |
| $SO_4$ µmol/l | 55164 | 916 | 1343 | 999 | 895 | 531 | 135 |
| Na µmol/l | 154539 | 2540 | 3741 | 4741 | 4654 | 1662 | 557 |
| Cl µmol/l | n.a. | 1439 | 2200 | 2821 | 2933 | 1015 | 296 |
| Ca µmol/l | 1241 | 1347 | 599 | 439 | 482 | 1332 | 586 |
| Mg µmol/l | 34584 | 1399 | 1917 | 2292 | 2218 | 876 | 407 |
| K µmol/l | 4617 | 113 | 143 | 624 | 223 | 74 | 49 |
| B µmol/l | 11962 | 88 | 125 | 312 | 300 | 76 | 120 |

**Table 2.** AMS radiocarbon dates of three fossil *Radix* sp. shells and of two charcoal samples.

| Site/Sample ID | Material | $^{14}$C-age yr BP | Res. corrected age cal. yr BP | Laboratory ID |
|---|---|---|---|---|
| Chiao Ho 34-36 S | Radix shell | 8540 ± 40 | 2211 ± 55 | Poz-53269 |
| Chiao Ho 40-42 S | Radix shell | 8480 ± 40 | 2428 ± 85 | Poz-53271 |
| Chiao Ho 34-36 C | Charcoal | 2275 ± 30 | 2275 ± 30 | Poz-53314 |
| Chiao Ho 40-42 C | Charcoal | 2400 ± 30 | 2400 ± 35 | Poz-53315 |
| Nama Chu | Radix shell | 12670 ± 60 | 7393 ± 114 | Poz-53277 |



**Table 3.** Dose rate, equivalent dose and ESR age.

Table 3: Dose rate, equivalent dose and ESR age.

| | | | | | | | | |
|---|---|---|---|---|---|---|---|---|
| $^{238}$U (Bq/kg) | 27.6 | ± | 1.5 | Internal U (ppm)* | | 2.8 | ± | 2.7 |
| $^{226}$Ra (Bq/kg) | 14.6 | ± | 0.3 | Internal dose rate (Gy/ka), early uptake | | 0.66 | ± | 0.46 |
| $^{232}$Th (Bq/kg) | 18.4 | ± | 0.2 | Internal dose rate (Gy/ka), linear uptake | | 0.33 | ± | 0.33 |
| K (Bq/kg) | 687 | ± | 5 | Total dose rate (Gy/ka), early uptake | | 3.97 | ± | 0.49 |
| External dose rate (Gy/ka) | 2.91 | ± | 0.21 | Total dose rate (Gy/ka), linear uptake | | 3.64 | ± | 0.41 |
| Cosmic dose rate (Gy/ka) | 0.40 | ± | 0.04 | Age, early uptake (ka) | | 7.4 | ± | 1.0 |
| $D_e$ (Gy) | 29.5 | ± | 1.1 | Age, linear uptake (ka) | | 8.1 | ± | 1.0 |

    * mean value of Holocene shells (Schellmann et al., 2008).


**Table 4.** Classification, and biological and ecological traits of early Middle Holocene molluscs from
the study area using the best modern analogue approach. Compiled from Burky et al., 1981; Clewing
et al., 2013, 2014a, 2014b; Frömming, 1956; Gittenberger et al., 1998; Glöer, 2002; Killeen et al.,
2004; Meier-Brook, 1969, 1975; Økland and Kuiper, 1982; Økland, 1990; Taft et al., 2012, 2013;
Turner et al., 1998; Wilckens, 2014; Zettler et al., 2006; and personal observations.

| Taxon | *Radix* sp. | *Gyraulus* sp. | *Valvata* sp. | *Pisidium* sp. |
|---|---|---|---|---|
| Classification | Gastropoda, Basommatophora, Lymnaeidae | Gastropoda, Basommatophora, Planorbidae | Gastropoda, Allogastropoda, Valvatidae | Bivalvia, Veneroida, Sphaeriidae |
| Life span (years) | 0.5-1.5 | 1-1.5 | 1-2 | 0.5-3 |
| Salinity range | freshwater to meso-haline (≤ 14 psu) | freshwater to oligo-haline (≤ 5 psu) | freshwater to oligo-haline (≤ 5 psu) | freshwater to oligo-haline (≤ 3 psu) |
| pH range | 5.2-10.4 | 5.0-10.4 | 5.0-9.6 | 4.0-9.3 |
| Aquatic system | wetlands, fluvial and lacustrine systems (moderate water movement preferred) | fluvial and lacustrine systems (still water conditions preferred) | fluvial and lacustrine systems (still to slow moving water conditions preferred) | wetlands, fluvial and lacustrine systems (moderate water movement preferred) |
| Water depth | most common in shallow littoral (ca. 0.1-2 m) | most common in shallow littoral (ca. 0.1-2 m) | most common in littoral (1.5-3 m) | most common in shallow littoral (ca. 0.1-2 m) |
| Substrate | epibenthic on all kinds of substrates (e.g. pebbles, sand, gyttja, water plants) | epibenthic on different solid substrates (e.g. pebbles, water plants) and on gyttja | epibenthic on all kinds of substrates (preferably organic-rich sediment) | endo- or epibenthic; soft substrates (most common in/on organic-rich silt and fine sand) |





**Table 5.** Size parameters and number of sub-samples of *Radix* shells used for stable isotope analyses.

| Sample ID | Height in cm | N whorls | N sub-samples |
|---|---|---|---|
| NC1 | 1.59 | 4.6 | 38 |
| NC2 | 1.34 | 5.1 | 34 |
| NC3 | 1.39 | 5 | 29 |
| NC4 | 1.49 | 4.8 | 30 |
| NC5 | 1.28 | 3.6 | 21 |


**Table 6.** $\delta^{13}$C and $\delta^{18}$O values from the five selected *Radix* shells NC1-5. Letter "a" indicates the sub-
sample from the outer rim of the aperture and thus the latest/youngest shell in ontogeny. The last
letters are mostly combined with numbers and vary due to the different sizes of the shells and
corresponding differences in maximum sub-sample numbers and represent the earliest (embryonic)
and thus oldest shell in ontogeny (z12, z8, z3, z4, u). Data are presented in individual graphs in Fig. 6.

| | NC1 | | NC2 | | NC3 | | NC4 | | NC5 | |
|---|---|---|---|---|---|---|---|---|---|---|
| | $\delta^{13}$C | $\delta^{18}$O | $\delta^{13}$C | $\delta^{18}$O | $\delta^{13}$C | $\delta^{18}$O | $\delta^{13}$C | $\delta^{18}$O | $\delta^{13}$C | $\delta^{18}$O |
| a | 8.0 | -6.6 | 6.4 | -7.0 | 6.0 | -8.4 | 6.8 | -7.9 | 7.8 | -4.4 |
| b | 7.8 | -4.4 | 6.9 | -6.0 | 5.1 | -8.4 | 5.9 | -8.1 | 7.7 | -4.1 |
| c | 6.7 | -5.8 | 6.5 | -7.4 | 4.6 | -8.4 | 6.2 | -8.1 | 8.3 | -3.2 |
| d | 6.5 | -6.6 | 6.2 | -7.9 | 5.0 | -8.0 | 6.2 | -7.9 | 8.1 | -2.5 |
| e | 6.3 | -7.0 | 6.2 | -8.1 | 5.1 | -7.9 | 6.0 | -7.8 | 7.7 | -3.6 |
| f | 5.8 | -7.6 | 6.2 | -8.3 | 4.9 | -8.4 | 5.8 | -7.8 | 7.6 | -4.3 |
| g | 5.7 | -8.1 | 6.2 | -8.3 | 4.9 | -8.7 | 5.8 | -7.9 | 7.6 | -4.4 |
| h | 5.6 | -8.0 | 5.7 | -8.5 | 4.8 | -9.2 | 5.8 | -7.7 | 7.7 | -4.5 |
| i | 5.2 | -8.1 | 5.7 | -8.7 | 4.7 | -9.4 | 5.4 | -7.6 | 7.6 | -4.9 |
| j | 5.1 | -8.2 | 5.6 | -8.7 | 4.6 | -9.7 | 5.5 | -7.6 | 6.1 | -5.3 |
| k | 4.7 | -8.3 | 5.4 | -8.7 | 4.4 | -9.8 | 5.3 | -7.8 | 5.6 | -5.3 |
| l | 4.9 | -8.2 | 5.1 | -8.8 | 3.9 | -10.0 | 4.8 | -7.8 | 4.8 | -5.5 |
| m | 5.3 | -8.1 | 4.6 | -8.7 | 4.2 | -10.2 | 4.6 | -7.6 | 4.8 | -5.4 |
| n | 4.6 | -8.3 | 4.4 | -8.8 | 5.1 | -9.9 | 3.7 | -7.9 | 5.1 | -5.2 |
| o | 4.9 | -8.5 | 4.4 | -6.5 | 4.6 | -10.2 | 3.5 | -7.8 | 5.4 | -5.1 |
| p | 5.0 | -8.4 | 4.6 | -8.9 | 4.8 | -10.1 | 3.3 | -7.9 | 5.6 | -4.6 |
| q | 5.2 | -8.1 | 4.3 | -9.1 | 4.3 | -10.0 | 3.2 | -7.7 | 5.2 | -4.7 |
| r | 5.0 | -8.4 | 4.1 | -9.1 | 4.6 | -9.8 | 3.6 | -7.6 | 4.9 | -5.4 |
| s | 5.3 | -7.9 | 4.6 | -9.0 | 4.9 | -9.3 | 3.8 | -7.7 | 5.4 | -5.0 |
| t | 5.9 | -7.6 | 4.0 | -9.0 | 5.7 | -8.9 | 4.1 | -7.7 | 7.2 | -4.1 |
| u | 5.7 | -7.3 | 4.0 | -8.6 | 6.5 | -8.2 | 4.5 | -7.5 | 6.4 | -4.9 |
| v | 5.8 | -6.9 | 4.6 | -8.1 | 6.2 | -9.0 | 5.3 | -7.3 | | |
| w | 8.4 | -6.8 | 5.6 | -8.2 | 6.2 | -9.0 | 6.5 | -6.9 | | |
| x | 8.1 | -6.4 | 5.4 | -7.6 | 5.2 | -10.1 | 6.8 | -6.7 | | |
| y | 6.6 | -7.0 | 4.8 | -7.2 | 4.5 | -9.1 | 6.9 | -6.2 | | |
| z | 6.0 | -7.6 | 4.4 | -7.9 | 4.0 | -9.4 | 6.3 | -6.2 | | |
| z1 | 5.7 | -7.6 | 4.3 | -8.4 | 4.2 | -9.5 | 6.6 | -7.0 | | |
| z2 | 5.3 | -7.4 | 4.0 | -9.1 | 3.7 | -9.9 | 7.4 | -7.3 | | |





| z3 | 5.6 | -7.1 | 6.6 | -8.5 | 4.0 | -9.8 | 6.4 | -7.6 | | |
|----|-----|------|-----|------|-----|------|-----|------|---|---|
| z4 | 5.8 | -7.6 | 4.9 | -9.0 | | | 6.2 | -7.5 | | |
| z5 | 6.6 | -7.6 | 4.0 | -9.3 | | | | | | |
| z6 | 5.6 | -7.6 | 3.5 | -9.3 | | | | | | |
| z7 | 5.4 | -8.1 | 4.1 | -9.4 | | | | | | |
| z8 | 5.3 | -7.4 | 4.2 | -8.5 | | | | | | |
| z9 | 4.9 | -7.9 | | | | | | | | |
| z10 | 4.6 | -7.8 | | | | | | | | |
| z11 | 5.3 | -7.2 | | | | | | | | |
| z12 | 5.5 | -8.0 | | | | | | | | |
| Ø | 5.8 | -7.5 | 5.1 | -8.4 | 4.9 | -9.3 | 5.4 | -7.5 | 6.5 | -4.6 |
