# Peer review of "Intra-seasonal hydrological processes on the western Tibetan Plateau: Monsoonal"

_Climate of the Past, 2019_

## Referee Comment (RC1) · Anonymous Referee #1 · 26 Apr 2019

Summary:

The authors investigate Mid-Holocene lake sediments of the Nyak Co lake side in an attempt use fossil evidence to infer intra-seasonal hydrological processes, identify and assess the significance of important moisture sources for the lake dynamics. The precipitation of the Nyak Co lake side is in proximity of the northern extent of the SW Asia summer monsoon, making the site potentially valuable in the detection of past changes of this extent. Five gastropod (Radix) shells provide most of the basis of the work, and they are subsampled to be used for temporally high resolution archives showing variability on a sub-annual scale. One of the authors' main conclusion is that

the northern extent of the Asian monsoon was different in the Mid-Holocene.

The manuscript presents new data potentially useful in addressing a number of hypotheses. The addressed problem is within the scope of CP. However, the data is insufficient in addressing the problem of changing monsoon precipitation in the Mid-Holocene, and related (rather substantial) conclusions cannot be made based on the presented results. Results and data seem overinterpreted. Not all scientific assumptions and methods to address this problem are clearly stated or valid. Scientific quality is further compromised by a lack of discussion of the large body of work related to monsoon or climate reconstruction of the Mid-Holocene. Presentation in text and figures do not allow the reader to easily follow. Given the serious flaws of this manuscript, I cannot recommend publication of this manuscript in its current form and with its intended focus, but I do believe the data are valuable in scientific discussion with different (and clearer) focus.

Specific comments:

Geographical and geological settings are well described. However, given the focus and the listed implications of the study, the introduction does not provide an adequate overview of the modern- and palaeo-climatology in the region. Most statements about it are vague and use confusing or inadequate descriptions. Furthermore, previous Mid-Holocene reconstruction efforts, such as PMIP simulations, are not mentioned at all. The description of monsoon dynamics that would allow more merited discussion of results and interpretations in that context are superficial. Furthermore, the goals laid out for the study in the introduction are very broad and beyond the scope of any single study. There is no mention of the specific hypotheses the authors seem to want to address. Consequently, the introduction does not sufficiently focus the reader on a problem.

The method section is on one hand lacking descriptions (e.g. computation of correlation coefficients and parts referring to software tools rather than the methods the

software tools apply). On the other hand, the processing steps for the samples are described in such detail that would be more merited in the supplemental material. It is unclear why this detail is needed in order to understand and discuss the results. However, I must acknowledge that my expertise does not cover such methods, so I am not able to truly assess the necessity of this level of detail.

The results section is difficult to read as it often does not highlight important results, but refers to tables instead. It does not feel like the reader is guided through the results in a manner that would also allow them to better understand the discussion that follows. These tables also include details that do not seem directly relevant to the problems the study addresses.

The reasoning in discussion is not easy to follow and insufficiently referenced when climate is discussed. In many cases, the authors seem to make interpretations without presenting sufficient evidence. Furthermore, the authors do not use existing studies, specifically studies reconstructing Mid-Holocene climate, to help them in their interpretation and put their results in context. The way it is presented, the main conclusions regarding climate, specifically the monsoon extent, are not supported by the results. In the introduction, the authors state they aim to separate moisture sources signals, but 1) it is unclear how they do this and 2) they do not take advantage of existing palaeoclimatological studies that would provide a more solid basis for such a discussion. Furthermore, by the very definition of climate, the question of a climatological changes in the monsoon is not one that could even potentially be tackled with the sample size used as a basis for this study.

The figures are insufficient in quality and quantity. Some labels are impossible to read, and the relevance of figures to the focus of the manuscript is not always immediately apparent. On the other hand, there are not enough figures to guide the reader through the study. It is difficult to get an overview of all information given in the text, and there are no suitable figures to help with this. For example, there is no figure to summarise 1) the climatic setting, which authors picked as their focus, 2) the distribution of modernand palaeo- oxygen isotope values, which would be important for the discussion, and 3) the different types of findings and interpretations that ultimately lead to the main conclusions.

The general structure of the manuscript is good. However, there are serious flaws within each section, and the manuscript lacks focus overall. It is often unclear why certain items are mentioned at specific points in the text. This makes it especially difficult to follow the authors' reasoning. It often reads as disconnected pieces of information the readers seem to be expected to piece together on their own. Given the broad audience and expertise of CP, more (concise) explanation and guidance is needed.

While the language only has few grammatical issues, it is often vague. Furthermore, the vague nature of the language used when discussing climate gives the impression that the authors are not accustomed to discussions with the climate community. If this is the case, I would recommend closer collaboration with the climate community.

While my expertise is (palaeo)climatology and I do have a background in palaeontology, I found it very difficult to follow parts of the manuscript. In some cases, I am not sure if the reasoning itself or simply the presentation of the reasoning was problematic. It would be easier to assess the scientific quality if the presentation quality were better.

Overall, I believe the authors have a valuable dataset that is potentially useful for testing a number of different hypotheses. However, the manuscript has serious flaws. Its major flaw is presentation and the attempt to tackle a climatological problem that 1) cannot be solved with only the data presented here, and 2) requires consideration of the abundant work of the (palaeo)climate community, which is almost completely ignored here. Due to those and aforementioned flaws, I cannot recommend publication of this manuscript in CP, but hope and suggest that the authors take full advantage of their valuable dataset by 1) deciding on one clear focus for the manuscript, and 2) addressing testable hypotheses either within their fields of expertise or in close collaboration with the climate community, and under consideration of the existing body of

work.

Technical specific comments:

L24: "Billions of people depend on ..." - please specify the nature of this dependence and state exactly what is meant by that. L38: "during" is not needed here. L53: In order to avoid confusion, when referring to monsoon characteristics. I suggest refraining from speaking about "atmospheric circulation patterns", a term with a more specific association in the climate community, L54: I presume the geographical extent is meant by "distribution". I suggest explicitly stating so. L56: The phrasing of the first part of the sentence is rather confusing. I suggest being more specific, e.g. "represents the source of water for large regions ...". L57-58: It is not clear what the authors are trying to say here. I presume the authors want to emphasise the ecological and societal importance of studying the climate and hydrology in the region. Please be more clear. L77: Another important factor to consider is the tectonic history of the study region. L79: Please also consider model-based reconstructions of palaeo moisture sources. L89-91: This sentence is confusing. I presume "was warmest and most humid" refers to only times within the Holocene, is talking about the Middle- (rather than Early) Holocene, and refers to to the Holocene Climatic Optimum. Please rephrase and clarify that. L96: What about differences in westerlies? L119-122: These are very broad questions. It is unclear what specific hypotheses are tested in this study. L227: If the data from the sites not marked on Fig. 2 are used in this study, please mark them on the figure instead of referring to another publication. It will help readers keep track of everything while reading the manuscript. L228: Please indicate clearly on the figure (using letters as reference or marking the outcrop in the legend) where this outcrop is. Is it B? L240: For interpretation and reproducibility, a mention of the actual method is far more valuable than mentioning the software, esp. when the software is not open and easily accessible to all. L260: This sentence is incomplete. "[. . .], but at Chiao Ho we did find such particles."? L260-263: Discussion of confidence are misplaced in the method section. L256-347: I am no expert in these methods and also cannot

judge the necessity of having this detailed information in the main manuscript rather than supplementary material. However, given that many of the details seem irrelevant for interpretations later and are not mentioned again, I believe much of it can probably moved to the supplement. L387: Please summarise your findings/interpretations of Mid-Holocene environments instead of simply referring to a table and letting the reader do that work, esp. since the tables contain more than only the information mentioned in/relevant to the text. L395: Again, a quick summary of results/highlights is necessary. Without this to guide the reader, the manuscript reads more like a data dump, and esp. readers from different scientific disciplines will have serious troubles with the manuscript without this sort of guidance. L406-407: Why are these correlations mentioned? How was this calculated? What kind of correlation measure was used? This is missing from the methods. L427-429: What is this assumption based on? L579-633: Why is this interpreted as monsoonal precipitation and changes? The reasoning is not clear and this interpretation is made without any discussion of palaeoclimate reconstructions, such as modelling efforts, which exist and can provide context for such discussion. L642-644: See above. Furthermore, 5 samples, corresponding to 5 different years, are not sufficient to make any significant interpretations of monsoon dynamics. L682: Even if one could confidently attribute measured signals to monsoonal precipitation, the sample size of the data used as a basis for this study would not allow general statements about such differences in climate at the time.

---

## Short Comment (SC1) · 3 May 2019

Dear anonymous reviewer, thank you for commenting on our manuscript. In your introductive summary you write that "one of the authors' main conclusion is that the northern extent of the Asian monsoon was different in the Mid-Holocene". This is not correct. In our conclusions we write that "the northern boundary of the SW Asian summer monsoon was in a similar position as in modern times" (lines 683-684). Still in the summary you write "the data is insufficient in addressing the problem of changing monsoon precipitation in the Mid-Holocene . . ." It is not the scope of the paper to address changing monsoon precipitation during the Mid-Holocene but to look at the

seasonality during a short Mid-Holocene period. The scope is addressed in the title and the aim is outlined in the introduction (lines 118-122), saying that we attempt in differentiating moisture sources, particularly between monsoonal, regional convective and westerly-derived moisture. Also in the summary you write "results and data seem overinterpreted", however under your specific comments section we could not find any example what you exactly mean and what alternative interpretations you suggest. Under "specific comments" you criticize that we do not mention "specific hypotheses" to be addressed. In the introduction we write about the controversial discussion and outline our aim (lines 118-122) to figure out whether we can differentiate moisture sources with the intra-seasonal climate and weather archive Radix. This is a very clear focus. You doubt that it is necessary to describe the processing steps of the samples in such a detail. We believe it is very essential to be absolutely transparent here because it is not simply about sub-sampling a shell but you need to apply the right technique to sub-sample only the primary shell layer which is formed in the temporal resolution of a weather event during the life of the gastropod. On the other hand, secondary or tertiary shell layers may have formed much later and thus do not provide a high-resolution signal. Details are similarly important for the other methods. You mention that tables "also include details that do not seem directly relevant to the problems the study addresses". We believe that all the tables are directly relevant for the discussion. We appreciate if you can give us a more specific example of your concern. Regarding our aim to differentiate between moisture sources, you write "it is unclear how they do this". From line 512 to line 560 we present 6 paragraphs (a-f) explaining on which considerations the interpretation of isotopic signatures in the shells is based. This part of the manuscript we wrote particularly for those readers who are not familiar with isotope fractionation processes, also against the background that Climate of the Past has a broad readership. You write "they do not take advantage of existing palaeo-climatological studies" and later that we "ignore" existing literature. It is not our intention to ignore literature. The scope of our paper is not to review the existing literature about Mid-Holocene climate dynamics of the Tibetan Plateau, but specifically address intra-seasonal climate

(or weather) signals on the western Tibetan Plateau. We are not aware of other studies than those we have referenced dealing with intra-seasonal signals in that area. We will be grateful to you if you give us a specific example of which literature exactly we do ignore. You write "figures are insufficient in quality and quantity". We admit that you need to zoom into Fig.6 to easier read the labels. We will change this and put the 5 graphs separately. All other figures have a good to very good quality. We do not follow your argument that we need more figures. You state "that the authors are not accustomed to discussions with the climate community". We have authored dozens of palaeo-climate studies in journals such as Quaternary Science Reviews, Scientific Reports or Palaeo3 and reject your statement. In several places you write that the manuscript has "serious flaws" but unfortunately in a quite general way. It is thus difficult to reply to this and we would be grateful if you go into a deeper discussion with us. With kind regards Linda Taft

---

## Referee Comment (RC2) · Anonymous Referee #2 · 16 May 2019

Taft et al present a novel source of information about past monsoon variability - variations in oxygen and carbon isotopes from gastropod shells. Unlike other proxies, which average over decades or centuries, this proxy has the potential to provide subseasonal resolution about the monsoon. My comments fall into two general categories: improving the paper's organization and clarity, and improving the interpretations of climate from isotopes.

General comments on paper organization and clarity:

a) Some sections could be shortened to improve readability. Examples of places to reduce the details and make the main points clearer are: lines 89-117 (focus on summa-

rizing regional patterns rather than providing description of lots of lakes individually), lines 144-193 (focus more on details relevant to this study), lines 468-500 (combine with similar information in section 4.3.2), lines 562-633 (it is unhelpful to provide one entire paragraph for each shell; one paragraph summarizing the main similarities and differences would be preferable).

b) Are Table 4, Section 3.2, and Section 4.2 regarding mollusk ecological traits necessary? They don't seem to contribute to the main goal of the paper.

c) Much of the discussion is more appropriate for a results section. For example, section 4.1 could be added to section 3.1. Lines 562-633 could be moved to section 3.3. The main things that should remain in the discussion are the inferences about climate.

d) The paper would benefit from editing throughout for proper English usage.

Climatic interpretations of isotopes:

a) The d18O and the d13C proxies are very complex with multiple competing influences, as the authors describe on lines 512-560. Thus, there are many different ways to explain a particular isotope excursion. Having both d18O and d13C does not necessarily help, either, because d13C is so complex. For this reason, some of the detailed interpretations presented on lines 562-633 regarding certain excursions being due to soil inwash, for example, or others being due to meltwater pulses, etc., seem very arbitrary and overinterpreted. For one more specific example, the authors generally consider periods of low d18O variation to be ice periods, but on line 571-573 a similar low-variability period is considered "too long" to be due to ice and is assigned another cause (even though it is impossible to say anything definitive about how many weeks or months a particular part of the shell spans). A simpler, more defensible, and more objective approach might be to report on several relevant metrics (like mean, standard deviation, and range) and compare how these vary from modern to Holocene.

b) I wished for more isotopic data from modern shells to compare with the five Holocene shells, in order to more quantitatively describe the modern-Holocene differences. At one point, the authors give the mean of two modern shells and this is useful for a very first order comparison of the hydrology (but one that could also be explained by changes in lake water residence time that we already know about between early/mid Holocene and modern). The stated goals of the paper are to look at more of the sub-seasonal signal, though, and for that we really need to compare with sub-seasonal signals of modern samples. Perhaps the authors have published such data in other papers. In that case, it would be useful to present it here again for comparison.

c) Given the large interannual variability in the monsoon region, it is unclear that 5 years is enough to truly give a good sample of Holocene monsoon climate. This is unfortunate, because I know how much work goes into sub-seasonally sampling even one shell! But it is important to recognize what these results do and don't tell us.

d) I was confused about the conclusion that the precipitation is not continuous, but in pulses. Generally, precipitation does occur in pulses (storms), even in locations within the core monsoon. Particularly in this dry part of the world, it doesn't rain every day, but certain weather systems will deliver moisture from time to time. So, this conclusion seemed obvious and non-consequential. The authors mentioned that lakes on the eastern Tibetan Plateau reveal single extended events (line 640-641), but there are other factors such as significant groundwater inflow that could smooth a d18O series.

e) It was also unclear how the conclusions about the northern boundary of the monsoon were reached. How can you tell this from one lake? You just know that this particular lake received monsoon moisture both today and in the early/mid Holocene. I think it is very hard to say, based on difficult-to-interpret isotope data and only five years worth of data, that this lake received more monsoon rainfall during the Holocene than today. It seems like a transect would be needed to really answer this question.

f) Conclusions distinguishing monsoon precipitation from meltwater influence do not

seem supportable because the isotopic ranges for these sources are not obviously differentiable (according to lines 512-527, estimates for monsoon precipitation and snowmelt are both around -14 per mil).
* * *

---

## Author Comment (AC1) · 22 May 2019

Taft et al present a novel source of information about past monsoon variability - variations in oxygen and carbon isotopes from gastropod shells. Unlike other proxies, which average over decades or centuries, this proxy has the potential to provide subseasonal resolution about the monsoon. My comments fall into two general categories: improving the paper's organization and clarity, and improving the interpretations of climate from isotopes.

General comments on paper organization and clarity: a) Some sections could be shortened to improve readability. Examples of places to reduce the details and make the
main points clearer are: lines 89-117 (focus on summarizing regional patterns rather than providing description of lots of lakes individually),

Our weather and climate archive Radix lives in lakes and thus we preferably address former lake studies from four lakes of the western Tibetan Plateau in which the authors have presented data we consider relevant for the interpretation of our results. These references show that there are some similarities across the region and we agree that we should better emphasize these (and we will try so), but on the other hand it also shows how differently the lakes developed. Tso Moriri and Tso Kar e.g. are located in neighboring valleys. While Tso Kar was about a hundred meters deep during the Mid-Holocene, it is a shallow pond nowadays. On the other hand Tso Moriri is still ca. 100 m deep and the water level has dropped for a couple of meters only since the Mid-Holocene. This is due to the different drainage basins which is much larger in the case of Tso Moriri and the elevation of the surrounding mountains is also higher there, etc. On the other hand Tso Kar was stronger influenced by tectonics. We will rephrase this paragraph to make clearer why we mention these studies and try to shorten the text.

lines 144-193 (focus more on details relevant to this study),

We consider the information on the hydromorphology of the drainage basin and other details of the study area important in order to better understand regional processes potentially affecting isotope values. It makes e.g. a difference if there are glaciers in the catchment or not, etc. The size of the lake area e.g. provides information about potential moisture recycling, etc. We will shorten this paragraph and put a stronger focus.

lines 468-500 (combine with similar information in section 4.3.2),

These two paragraphs provide different information. The mean values average over the life-spans of the gastropods and primarily do not reflect seasonal changes while the sclerochronological isotope patterns do so. We believe the results are more transparent as it is arranged but will make an attempt to integrate these different categories.

lines 562-633 (it is unhelpful to provide one entire paragraph for each shell; one paragraph summarizing the main similarities and differences would be preferable).

We believe that it is essential to address each shell separately. Otherwise the reader may not be able to follow our reasoning. We understand from your comments that it is anyway not easy to follow the discussion and that you consider over-interpretation in parts. You also suggest that presentation of published data (see below) may help to better understand our conclusions. Against this background, we prefer not to shorten the discussion but make it even more transparent by adding information from modern shells as outlined under the paragraph below, "Climatic interpretations of isotopes: b)"

b) Are Table 4, Section 3.2, and Section 4.2 regarding mollusk ecological traits necessary? They don't seem to contribute to the main goal of the paper.

This table is important because the interpretation of isotope patterns to some degree depend on the habitat situation of Mid-Holocene Radix. The data confirms on the one hand that the nowadays saline pond was a freshwater system and thus the water residence time was relatively short and on the other hand that Radix lived in shallow water which is important regarding the signal strength. If it lived several meters deeper some buffering effects of the hydro-climate signals have to be considered. If you do not analyze the mollusk assemblage but take the ecology of Radix alone it also could have lived in a mesohaline environment and in deeper water. We will rephrase section 4.2. to make our point clearer.

c) Much of the discussion is more appropriate for a results section. For example, section 4.1 could be added to section 3.1. Lines 562-633 could be moved to section 3.3. The main things that should remain in the discussion are the inferences about climate.

We can do this, however we followed the concept of not mixing results and interpretation and need the comment of the editor here.

d) The paper would benefit from editing throughout for proper English usage.

As some of us have lived in the USA for several years, we were confident that our English usage is sufficient. We will ask a native speaker to edit the manuscript.

Climatic interpretations of isotopes: a) The d18O and the d13C proxies are very complex with multiple competing influences, as the authors describe on lines 512-560. Thus, there are many different ways to explain a particular isotope excursion. Having both d18O and d13C does not necessarily help, either, because d13C is so complex. For this reason, some of the detailed interpretations presented on lines 562-633 regarding certain excursions being due to soil inwash, for example, or others being due to meltwater pulses, etc., seem very arbitrary and overinterpreted. For one more specific example, the authors generally consider periods of low d18O variation to be ice periods, but on line 571-573 a similar low-variability period is considered "too long" to be due to ice and is assigned another cause (even though it is impossible to say anything definitive about how many weeks or months a particular part of the shell spans). A simpler, more defensible, and more objective approach might be to report on several relevant metrics (like mean, standard deviation, and range) and compare how these vary from modern to Holocene.

We believe we can overcome these problems by following your suggestion (see below) to include published data of modern Radix shells from several lake and climate settings (Taft et al. 2012, 2013). It will become evident then that e.g. inwash of carbon dissolved in soil is not an over-interpretation but there is a clear correlation of negative d13C excursions and stronger (monsoonal) rainfall. The modern shells were collected by ourselves and therefore we knew quite exactly the life-spans and could relate it to synoptical data of particular years allowing interpretations such as ice cover period, etc. We did not want to overload the paper with already published data on modern shells but we see your points and follow your idea to include these. It certainly will

make it easier for the reader to judge on our interpretation of Mid-Holocene isotope patterns.

b) I wished for more isotopic data from modern shells to compare with the five Holocene shells, in order to more quantitatively describe the modern-Holocene differences. At one point, the authors give the mean of two modern shells and this is useful for a very first order comparison of the hydrology (but one that could also be explained by changes in lake water residence time that we already know about between early/mid Holocene and modern). The stated goals of the paper are to look at more of the subseasonal signal, though, and for that we really need to compare with sub-seasonal signals of modern samples. Perhaps the authors have published such data in other papers. In that case, it would be useful to present it here again for comparison.

The Nama Chu pond is too saline nowadays that Radix can live there. Therefore we could not apply a direct modern analogue but used two modern shells from neighboring Nyak Co (eastern Bangong Co). It is a very good idea to present selected published data again. Particularly because most of the published isotope data are from modern shells and thus could be related to synoptical data, it will likely help to better demonstrate the potential of the archive.

c) Given the large interannual variability in the monsoon region, it is unclear that 5 years is enough to truly give a good sample of Holocene monsoon climate. This is unfortunate, because I know how much work goes into sub-seasonally sampling even one shell! But it is important to recognize what these results do and don't tell us.

Papers have been published in high-ranked journals which present one or two sclerochronological isotope patterns in order to open some weather window of the past. You are completely right that more is mostly better but we believe that we found a good balance and that the data presented allows us to make the conclusions we did. Again, this will hopefully become clearer when we have included isotope patterns of modern shells from the Tibetan Plateau.

d) I was confused about the conclusion that the precipitation is not continuous, but in pulses. Generally, precipitation does occur in pulses (storms), even in locations within the core monsoon. Particularly in this dry part of the world, it doesn't rain every day, but certain weather systems will deliver moisture from time to time. So, this conclusion seemed obvious and non-consequential. The authors mentioned that lakes on the eastern Tibetan Plateau reveal single extended events (line 640-641), but there are other factors such as significant groundwater inflow that could smooth a d18O series.

With pulses we do not address single weather events but rather a sequence of cloud bands which rained out over the research area during a certain period. This you may either call a short monsoon season (single "pulse") or a double-peak monsoon season. We assume that between two such periods ("pulses") the cloud bands rained out further south or the rain was too weak a signal to be archived in the shells. Under modern climate conditions, with Bangong Co located at the northern limit of monsoon moisture, rain periods strong enough to be archived in the shells may occur only exceptionally (Taft et al. 2013). In the Mid-Holocene shells, however, we could identify monsoonal rainfall during two periods which means that the rainfall was significantly stronger then. The already published data e.g. on the eastern plateau lakes Bangda Co and Donggi Cona will be presented here again (following your suggestion). These graphs show that although the monsoonal rainfall derives from many single weather events there is an overlying pattern defining a single monsoon season (this is not in "pulses"). The rate of summer growth of Radix shells allows for a maximal resolution of ca. 1 week (for temporal resolution see f) from lines 556-560) allowing some averaging of storm events. The isotopic signal of the rainfall also depends on the humidity. It differs strongly between early and late monsoon season e.g. (Taft et al. 2012,2013). You are right that other factors than rainfall such as groundwater inflow can influence the the isotopic pattern. At this elevation groundwater derives either from rainfall (thus providing a good average rain signal) because the permafrost does not allow the water to penetrate to deeper soil or rock layers and on the other hand thawing of permafrost itself produces groundwater. Regarding our interpretation of isotope patterns we are fully aware that

several hydrological processes interact. Our interpretations however put the focus on the dominant factor likely superimposing others. We suggest to rephrase these parts of the manuscript that the reader can easier follow our interpretation.

e) It was also unclear how the conclusions about the northern boundary of the monsoon were reached. How can you tell this from one lake? You just know that this particular lake received monsoon moisture both today and in the early/mid Holocene. I think it is very hard to say, based on difficult-to-interpret isotope data and only five years worth of data, that this lake received more monsoon rainfall during the Holocene than today. It seems like a transect would be needed to really answer this question.

We could find in modern Radix shells from the Tibetan Plateau (Taft et al. 2012, 2013) that weak rainfall cannot be identified and is not archived respectively. Drizzle e.g. is influencing the humidity but is too little an amount to create a signal in the shell of a gastropod which lives in water. The inwash of soil with dissolved terrestrial carbon is also possible only by strong rain (see also comment above about relation of rainfall and soil inwash identified in isotope patterns). We cannot tell however how strong it exactly has to be. It may vary from system to system too. We can see a clear difference between modern and Mid-Holocene isotope shell patterns though. From this we conclude that the monsoonal rainfall was stronger during the Mid-Holocene.

f) Conclusions distinguishing monsoon precipitation from meltwater influence do not seem supportable because the isotopic ranges for these sources are not obviously differentiable (according to lines 512-527, estimates for monsoon precipitation and snowmelt are both around -14 per mil).

We do not identify the snowmelt solely by d18O per mil but in the context of the (sub-) seasons archived in the shell. Ice cover is nowadays from November to April. Snowmelt becomes significant from May and peaks in July. The general pattern was likely similar during the Mid-Holocene. We assume that it can be ruled out that the monsoonal rainfall has reached the area directly after the ice cover period in May but

that the meltwater was dominant in May. Convective rainfall could (and did) occur already in May when evaporation of moisture from thawing soil and open lake surfaces became effective. Later in the season the meltwater inflow was superimposed by increasing (insolation) evaporation. Monsoonal rainfall with d18O similar to meltwater likely did not reach the area before July. From this seasonal chronology we believe we can differentiate between the two moisture sources although the d18O values were similar. Additionally, however, we argue that "stronger" monsoonal rainfall triggers inwash of terrestrial carbon and we thus can see d13C excursions to more negative values approximately synchronously to d18O excursions. This negative d13C peak does not occur during May meltwater inflow.

Please also note the supplement to this comment:
https://www.clim-past-discuss.net/cp-2019-23/cp-2019-23-AC1-supplement.pdf